# Extinction and Optical Depth Retrievals for CALIPSO's Version 4 Data Release

Stuart A. Young[1], Mark A. Vaughan[2], Anne Garnier[1], Jason L. Tackett[1], James D. Lambeth[1], Kathleen A. Powell[2]

[1]Science Systems and Applications, Inc. (SSAI), Hampton, VA 23666, USA
[2]NASA Langley Research Center, Hampton, VA 23681, USA

*Correspondence to*: Stuart A. Young (stuart.young01@gmail.com)

**Abstract.** The Cloud-Aerosol Lidar with Orthogonal Polarization (CALIOP) on board the Cloud-Aerosol Lidar Infrared Pathfinder Satellite Observations (CALIPSO) satellite has been making near-global height-resolved measurements of cloud

and aerosol layers since mid-June 2006. Version 4.10 (V4) of the CALIOP data products, released in November, 2016, introduces extensive upgrades to the algorithms used to retrieve the spatial and optical properties of these layers, and thus there are both obvious and subtle differences between V4 and previous data releases. This paper describes the improvements made to the extinction retrieval algorithms and illustrates the impacts of these changes on the extinction and optical depth estimates reported in the CALIPSO lidar level 2 data products. The lidar ratios for both aerosols and ice clouds are generally higher than

in previous data releases, resulting in generally higher extinction coefficients and optical depths in V4. A newly implemented algorithm for retrieving extinction coefficients in opaque layers is described and its impact examined. Precise lidar ratio estimates are also retrieved in these opaque layers. For semi-transparent cirrus clouds, comparisons between CALIOP V4 optical depths and the optical depths reported by MODIS collection 6 show substantial improvements relative to earlier comparisons between CALIOP version 3 and MODIS collection 5.

## 20 1 Introduction

The Cloud-Aerosol Lidar with Orthogonal Polarization (CALIOP) lidar (Hunt et al., 2009) on board the Cloud-Aerosol Lidar Infrared Pathfinder Satellite Observations (CALIPSO) satellite (Winker et al., 2010) has now acquired over 12 years of global, height-resolved information on the location, extent and optical properties of clouds and aerosols between latitudes of 82 °S and 82 °N. The process of extracting the desired optical and physical properties from the raw measurements is accomplished

by a sophisticated suite of retrieval algorithms that performs a range of functions. These include calibration, determining the horizontal and vertical extent of atmospheric regions that contain particulate (cloud or aerosol) matter, which we call features (Young and Vaughan, 2009, henceforth YV09), identifying the types of particles in those features, and retrieving profiles and layer integrals of the particulate backscatter and extinction in those features. An overview of the CALIOP processing algorithms has been provided by Winker et al. (2009).

The tasks accomplished by the extinction algorithms include the retrieval of all of the particulate optical profiles, including backscatter coefficients, extinction coefficients and their layer integrals (i.e., optical depths), particulate depolarization ratios and particulate color ratios (the ratio of the particulate backscatter coefficients at 1064 nm and 532 nm). The extinction algorithm requires initial estimates of the particulate extinction-to-backscatter ratios, called lidar ratios, to retrieve extinction

and optical depth from the measured attenuated backscatter. These lidar ratios vary depending on the type of particle (ice cloud, water cloud, or aerosol subtype), as determined by the preceding algorithms. Whenever relevant, contributions from multiple scattering are taken into account through a multiple scattering factor (YV09). The initially assigned lidar ratios and multiple scattering factors, called analysis parameters, are often critically important drivers of the retrieved extinction and optical depth. A notable exception is when the retrievals can be constrained by estimates of the feature two-way transmittance,

which can be directly derived from CALIOP attenuated backscatter measurements made immediately above and below the boundaries of semi-transparent features (YV09). This procedure, already implemented in version 3 (V3), has been extended in V4. In case of opaque layers, the initial lidar ratio in V4 is estimated from measurements rather than being assigned based on layer type, as in V3 (Sect. 2.2.3). This important change is paired with a totally new dedicated surface detection algorithm (Vaughan et al., 2018), which leads to a significantly more reliable identification of opaque layers than in V3. When the

retrievals cannot be constrained or when the initial lidar ratio cannot be determined directly from the measurements, the extinction algorithm uses a set of initial lidar ratios, many of which have been significantly changed in V4.

In addition to changes to the extinction retrieval algorithms and the analysis parameters, significant upgrades to preceding algorithms in the analysis chain also affect the retrieved quantities. For example, the new 532 nm nighttime calibration algorithm (Kar et al., 2018) normalizes the measured attenuated backscatter profiles to a molecular model at higher altitudes

than in the previous versions in order to avoid errors caused by the presence of aerosols in the supposedly aerosol-free calibration region. This can lead to significant changes to the attenuated backscatter, particularly in the tropics, and would be expected, generally, to cause increases in the retrieved optical quantities. The number and vertical and horizontal extent of the features subsequently detected have been impacted by changes in the calibration procedures (Kar et al. 2018; Getzewich et al., 2018) and the use of an improved molecular model (Kar et al., 2018). The retrieval of optical property profiles has also been

affected by upgrades to the cloud-aerosol discrimination (CAD) algorithm (Liu et al., 2018).  Large changes in calibration, especially at 1064 nm (Vaughan et al, 2018a), necessitated a complete restructuring of the probability density functions that drive the CAD algorithm, and hence the classification of features as clouds or aerosols is not uniformly consistent from V3 to V4.  Additionally, the generic stratospheric layer type identified in V3 has been eliminated in V4, and the CAD algorithm is now applied to all features detected in both the troposphere and the stratosphere. The algorithm that classifies clouds as water

or ice has been refined in V4 (Avery et al., 2018). Similarly, the aerosol subtyping algorithms have been extensively revised, and the characteristic lidar ratios for several aerosol subtypes have been updated to reflect on-going research carried out over the past decade. V4 now includes a new tropospheric aerosol sub-type (dusty marine) and several stratospheric aerosol sub-types (Kim et al., 2018).

Although changes to the algorithms that precede the extinction retrieval algorithm can be expected to impact the extinction retrieval, this paper concentrates on the description of changes to the extinction algorithm and its associated analysis parameters. Details of the extinction and optical depth changes attributed to the upstream algorithms will be published separately (e.g., Kim et al., 2018).

The outline of this paper is as follows. Section 2 describes changes to the extinction retrieval process, beginning with a description of changes to the analysis parameters (the lidar ratios and multiple scattering factors for clouds and aerosols), then proceeds to a description of changes to the various components of the actual extinction retrieval algorithm, which includes the description of a new algorithm for retrievals in opaque layers. Section 2 also describes revised algorithms for adjusting lidar ratios to achieve successful solutions, modification of lidar ratio uncertainties, and changes to the extinction quality control

(QC) flags, which report the status of the extinction solution at the conclusion of the retrieval process. Section 3 presents the results of these changes and includes comparisons of the V4 and V3 retrievals in tropical clouds, lidar ratios retrieved in ice and water clouds using the new opaque layer algorithm, optical depths in opaque and semi-transparent clouds, and median profiles of extinction coefficient as a function of mid-cloud temperature and extinction QC value. We also provide advice on the use of extinction QC values to filter data, and end with some general caveats regarding the retrieval of extinction from

elastic backscatter lidars. The mathematical details of the changes presented in Sect. 2 are described in supplementary material. This supplementary material also includes descriptions of new criteria for determining the existence of solutions, an updated uncertainty analysis and details of new initialization to the backscatter solution and new initialization of the lidar ratio in opaque layers.

## 2 Changes in the retrieval process

The algorithms for retrieving profiles of particulate backscatter, extinction and optical depth are described in YV09 and Young et al. (2013, 2016), henceforth Y1316. Changes and new features in the analysis parameters and in the algorithms used in the V4 retrievals have led to substantial improvements in these products when compared with those reported in V3. These changes are outlined in the following sections, with the mathematical details appearing in the supplementary material. The most significant changes in the analysis parameters are to the initial estimates of particulate lidar ratios, and their uncertainties, for

both aerosols and ice clouds (Sect. 2.1). The multiple scattering factors used for retrieving extinction profiles in ice clouds and in opaque water clouds have also been changed.

The extinction algorithm has also been modified to significantly increase the number of constrained retrievals that are attempted, and the extensive changes made in the treatment of opaque layers now produce much improved extinction profiles in these features (Sect. 2.2). The iterative retrieval of the particulate backscatter at each point in the profile is now initialized

in a new way that ensures that the initial value is close to the final value with the combined benefits of reliable and rapid

convergence of the iteration. Further modifications to the retrieval algorithms include changes to the assessment of what is and what is not a successful retrieval, and these are reflected in revisions to the extinction QC flags.

### 2.1 Changes to the initial lidar ratios and multiple scattering factors

In this section we summarize the changes introduced in V4 to the initial lidar ratios for aerosols and clouds. These initial values
are used by the extinction retrieval algorithm for the analysis of semi-transparent layers where the retrievals cannot be constrained by an independent measurement of the effective two-way transmittance of the layer. (The effective, two-way layer transmittance, $T_P^2$, can be expressed as $\exp(-2\eta\tau_P)$ where $\eta\tau_P$ is the effective particulate optical depth of the layer, $\tau_P$ is the single-scattering optical depth, and $\eta$ is a layer-effective multiple scattering factor ($0 \leq \eta \leq 1$) that accounts for the apparent reduction in optical depth caused by the backscatter of multiply-scattered photons into the receiver (Platt, 1973). The effective
transmittance is biased high with respect to the single-scattering ($\eta = 1$) value.) The final values of the lidar ratios may be reduced from the initial values if it is found that the initial values are too large to permit a solution. For opaque layers, the initial lidar ratios are now derived from the measurements, as described in Sect. 2.2.3.

The relative uncertainties in the initial lidar ratios have also been changed in V4. In most cases these are considerably smaller than in V3. As the lidar ratio uncertainty is usually the dominant factor determining the overall uncertainties in the retrieved
extinction data products, these latter uncertainties are expected to be smaller than in V3.

In addition to changes in the lidar ratios and their uncertainties, there are also changes to the multiple scattering factors for clouds. Whereas these factors were constants in V3, in V4 they are now functions of cloud 532 nm attenuated backscatter centroid temperature for ice clouds (Garnier et al., 2015) and are obtained from 532 nm depolarization measurements for opaque water clouds (Hu, 2007).

### 2.1.1 Initial lidar ratios for aerosols

The initial lidar ratios and their uncertainties for several of the aerosol subtypes have been revised for V4 (Kim et al., 2018). As can be seen in Table 1, V4 specifies larger 532 nm lidar ratios for dust (by 10 %), clean marine (by 15 %) and clean continental (by 51 %) aerosols. These higher lidar ratios, which are consistent with improved knowledge gained over the past decade (e.g., Tesche et al., 2009; Nisantzi et al., 2015; Liu et al., 2015; Papagiannopoulos et al., 2016; Haarig et al., 2017),
contribute to higher values in the retrieved backscatter, extinction and optical depths than were reported in V3. Our better understanding of lidar ratio variability led to significant reductions in the uncertainties ascribed to the lidar ratios for marine, desert dust and smoke aerosols, resulting in generally in lower uncertainties in the retrieved products for these aerosol types. As explained in Kim et al. (2018) and documented in the V4 CALIPSO Data Products Catalog (https://www-calipso.larc.nasa.gov/documents/dpc_index.php), the nomenclatures of the aerosol type 3 "polluted continental" and type 6
"smoke" were changed in version 4.1 to "polluted continental/smoke" and "elevated smoke" respectively. V4 also adds a new

"dusty marine" tropospheric subtype and five new stratospheric subtypes. The lidar ratio for the dusty marine mixture is significantly higher than that for clean marine, but significantly lower than that for polluted dust. (Aerosols identified as dusty marine in V4 were generally classified as polluted dust in V3.) Some aerosol lidar ratios at 1064 nm have also been changed, as shown in Table 1. Note that there was no cloud-aerosol classification in the stratosphere in V3 and that all stratospheric

5    layers were assigned an initial lidar ratio of 25 sr and a multiple scattering factor of 1.

Combined with changes in the CAD and aerosol subtyping algorithms that have led to changes in the identification of some aerosol types, these new lidar ratios yield extinction profiles and optical depths that can differ noticeably from those reported in V3. For aerosols, changes to optical depths and extinction profiles are largely the result of changes to the lidar ratios for the particular aerosol subtypes, and of changes in the algorithms that determine these subtypes, and are only rarely due to changes

10    to the extinction algorithm. Changes in aerosol optical depths, therefore, are discussed in a separate publication (Kim et al., 2018).

**Table 1: Initial lidar ratios and their uncertainties, in units of steradians, that characterize each of the aerosol subtypes reported in V3 and V4. V4 values that have changed with respect to V3 are in bold text.**

| Aerosol Type | Version 3 | | Version 4 | |
|---|---|---|---|---|
| *Tropospheric* | *532 nm (sr)* | *1064 nm (sr)* | *532 nm (sr)* | *1064 nm (sr)* |
| Clean marine | 20 ± 6 | 45 ± 23 | **23 ± 5** | **23 ± 5** |
| Dust | 40 ± 20 | 55 ± 17 | **44 ± 9** | **44 ± 13** |
| Polluted continental / smoke | 70 ± 25 | 30 ± 14 | 70 ± 25 | 30 ± 14 |
| Clean continental | 35 ± 16 | 30 ± 17 | **53 ± 24** | 30 ± 17 |
| Polluted dust | 55 ± 22 | 48 ± 24 | 55 ± 22 | 48 ± 24 |
| Elevated smoke | 70 ± 28 | 40 ± 24 | 70 ± **16** | **30 ± 18** |
| Dusty marine | N/A | N/A | **37 ± 15** | **37 ± 15** |
| *Stratospheric* | | | | |
| Polar stratospheric aerosol | N/A | N/A | 50 ± 20 | 25 ± 10 |
| Volcanic ash | N/A | N/A | 44 ± 9 | 44 ± 13 |
| Sulfate / other | N/A | N/A | 50 ± 18 | 30 ± 14 |
| Smoke (stratosphere) | N/A | N/A | 70 ± 16 | 30 ± 18 |
| Generic stratospheric layer | 25 ± 10 | 25 ± 10 | N/A | N/A |

## 2.1.2 Multiple scattering factors and initial lidar ratios for clouds

### 2.1.2.1 Ice clouds

In contrast to V3 and earlier data releases, where the multiple scattering factors for ice clouds were assigned a constant value of $\eta = 0.6$, ice cloud multiple scattering factors in V4 are approximated by a sigmoid function of the temperature at the centroid
of the 532 nm attenuated backscatter coefficients within the cloud. The sigmoid function used in the V4 retrievals was derived by reconciling three years (2008, 2010, 2013) of CALIOP effective two-way transmittance measurements $\left( T^2_{CALIOP} = \exp\left(-2\eta\tau_{CALIOP}\right) \right)$ in semi-transparent clouds with the collocated absorption optical depths ($\tau_{IIR}$) retrieved by the CALIPSO Infrared Imaging Radiometer (IIR) at 12.05 µm (Garnier et al., 2015). This analysis used only clouds measured at night and identified with high confidence as being composed of randomly oriented ice (ROI) (Hu et al., 2009). Furthermore,
to guarantee the highest quality retrievals of $\tau_{IIR}$, it was restricted to single-layer clouds measured over oceans. As seen in Fig. 8a in Garnier et al. (2015), the effective optical depths (i.e., $\eta\ \tau_{CALIOP}$) obtained from direct measurements of $T^2_{CALIOP}$ must be rescaled by a temperature-dependent multiple scattering factor to ensure the appropriate temperature-dependent correspondence between $\tau_{CALIOP}$ and $\tau_{IIR}$. As shown in Fig. 1a, the V4 sigmoid approximation of $\eta$, which was retrieved assuming that these clouds are composed of severely roughened aggregated columns (Yang et al., 2013), increases from a
value of 0.46 at a centroid temperature of 0 °C to 0.76 at -90 °C.

As a consequence of the changes in the $\eta$ parameterization, the initial lidar ratio estimates for ice clouds have also been changed. In V3 and earlier data releases, an initial lidar ratio of 25 sr was used in all ice cloud extinction retrievals. Comparisons of V3 optical depths for semi-transparent ice clouds with IIR absorption optical depths at 12.05 µm (Garnier et al., 2015) and a radiative closure experiment using MODIS 11µm radiances by Holz et al. (2016) showed, on average, quite good agreement
for constrained retrievals. For unconstrained retrievals, however, several studies of semi-transparent ice clouds showed that, when used in conjunction with $\eta \approx 0.6$, the initial lidar ratio of 25 sr was generally too small. Josset et al. (2012) found, from their analyses of optical depths determined from CALIOP and CloudSat echoes from the ocean's surface, that the initial lidar ratio was about 25 % too low. Their analysis of semi-transparent cirrus produced a mean multiple scattering factor of 0.61 ± 0.15, with a corresponding mean lidar ratio of 33 ± 5 sr that showed slight variations as a function of temperature and latitude.
A similar conclusion was reached by Garnier et al. (2012), who compared CALIOP optical depths with those measured by the IIR at 12.05 µm and found that there was an inconsistency between CALIOP V3 constrained and unconstrained optical depths, and that using a revised initial lidar ratio derived from constrained retrievals greatly improved the comparisons. Holz et al. (2016) showed that the CALIOP unconstrained optical depths were substantially smaller than the MODIS Collection 5 values. The authors showed that a much better agreement was achieved when CALIOP unconstrained retrievals using a lidar ratio of
32 sr and a multiple scattering factor of 0.6 were compared with the MODIS Collection 6 (C6) optical depths.

To ensure consistency between unconstrained and constrained retrievals in semi-transparent ice clouds, the V4 initial lidar ratios have been derived from a statistical analysis of nighttime constrained retrievals in clouds identified with high confidence as being composed of ROI. These constrained retrievals used direct measurements of effective two-way transmittance, together with multiple scattering factors generated by the aforementioned sigmoid approximation function, to retrieve optimized estimates of the V4 initial cirrus cloud lidar ratios. As seen in Fig. 1, the resulting initial cirrus lidar ratios and the corresponding layer-effective lidar ratios (i.e., the product of lidar ratio and multiple scattering factor) both vary as functions of the layer attenuated backscatter centroid temperature. Like the multiple scattering factors, the initial cirrus lidar ratios (Fig. 1b) are approximated by a sigmoid function of backscatter centroid temperature, and decrease from a value of ~ 35 sr to ~ 20 sr as the layer centroid temperature decreases from 0 °C to -90 °C. It can be seen that the V4 effective lidar ratios (Fig. 1c) are greater than the V3 value ($25 \times 0.6 = 15$ sr) at all values of centroid temperature while the V4 lidar ratios themselves are larger than the V3 value of 25 sr for all temperatures above -70 °C. Based on the improved understanding of the natural variability of lidar ratio obtained from this statistical analysis, the relative uncertainty assigned to the initial lidar ratios for semi-transparent ice clouds has been reduced from 40 % in V3 to 25 % in V4.

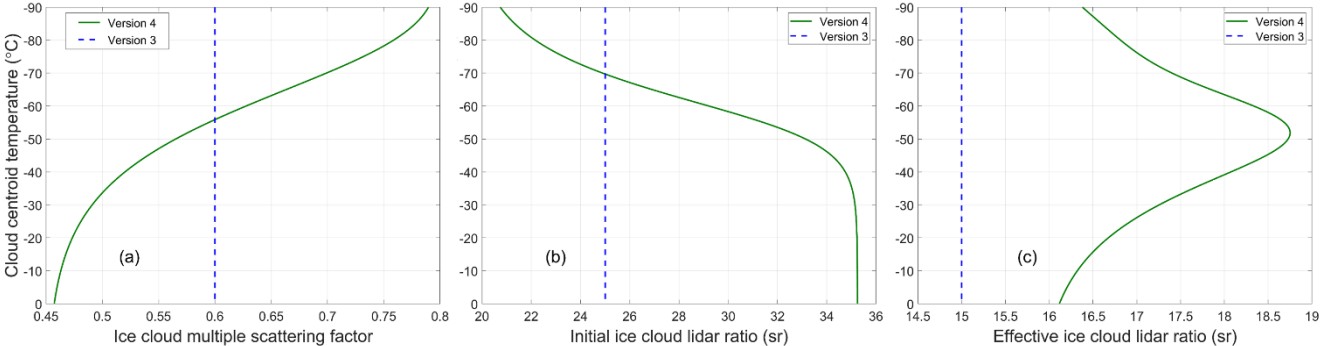

**Figure 1: CALIOP ice cloud scattering parameters for both V3 and V4 plotted as functions of cloud 532 nm attenuated backscatter centroid temperature, showing (a) multiple scattering factors, (b) lidar ratios, and (c) effective lidar ratios (i.e., multiple scattering factor × lidar ratio). The multiple scattering factors and lidar ratios are sigmoid approximations to measured values reported in Garnier et al. (2015).**

### 2.1.2.2 Water clouds

For semi-transparent water clouds, there have been no changes to either the initial lidar ratio or the multiple scattering factor. In V4 these have values of 19 sr and 0.6, respectively, as they did in V3. The relative uncertainty in the lidar ratio has, however, been reduced from 40 % to 15 %, reflecting substantial improvements in the cloud thermodynamic phase discrimination

algorithm (Avery et al., 2018) and the implementation of a new algorithm for estimating multiple scattering factors for opaque water clouds (Hu et al., 2007).

### 2.1.2.3 Clouds of unknown phase

In V3, when the cloud phase could not be identified, a constant lidar ratio of 22 sr was used, this being the average of the ice cloud (25 sr) and water cloud (19 sr) values. In V4, the ice and water lidar ratios are again averaged but now the ice-cloud lidar ratio is obtained from the sigmoid approximation function shown in Fig. 1b. Similarly, the multiple scattering factors in V4 are the average of the value for semi-transparent water clouds (0.6) and the sigmoid function values shown in Fig. 1a. In V3, a constant value of 0.6 was used.

### 2.2 Changes to the extinction retrieval algorithm

### 2.2.1 Increased number of constrained retrievals

Extinction profile retrievals that are constrained by measurements of the effective two-way transmittance of the layer are generally more accurate than those for which no constraint exists. Constrained retrievals generate optimized values of the layer lidar ratio that are precisely matched to the measured two-way transmittance. However, for unconstrained retrievals optimized adjustments of the lidar ratio are not possible (because no constraint is available), and hence the accuracy of the two-way transmittance estimates derived using this method depends critically on the accuracy of the initial lidar ratio. In V3, constrained retrievals were only performed when the estimated error in the retrieved lidar ratio was estimated to be less than 40 %. Garnier et al. (2015) found, however, good agreement between CALIOP constrained optical depths and IIR optical depths at smaller values of optical depths than implied by the 40 % limit on lidar ratio uncertainty. Indeed, low optical depths correspond to high effective two-way particulate transmittances and, hence, high lidar ratio uncertainties, as seen in Eq.7.4 of Liu et al. (2005). By removing the lidar ratio uncertainty threshold in V4, a far greater proportion of constrained retrievals is achieved. Note, however, that the optical depth uncertainty estimates for unconstrained retrievals depend directly on the uncertainties assigned to the initial lidar ratios. In contrast, the optical depth uncertainties for constrained retrievals depend only on the uncertainties in the measured data. The uncertainties in the lidar ratios obtained from constrained solutions likewise depend on the uncertainties in the measured optical depths and are entirely independent of the uncertainties associated with the initial lidar ratios. A paradoxical consequence of this dichotomy is that for features with large uncertainties in their measured optical depths, constrained solutions can generate larger uncertainties in the retrieved backscatter and extinction profiles and derived lidar ratios than would have been the case had an unconstrained retrieval been used instead.

Constrained retrievals use measurements of the effective two-way layer transmittance and its uncertainty that are determined by comparing signals from above and below the layer. The determination of the layer boundaries is detailed in Vaughan et al. (2005 and 2009). Briefly, the initial determination of cloud base is determined as that range at which the attenuated scattering ratio (the ratio of the normalized, range-corrected backscatter signal to a molecular backscatter model) drops below a range-

dependent threshold that is determined largely by SNR and signal attenuation by the overlying atmosphere. This initial estimate of cloud base is further refined by continuing to search below this altitude for a region where the attenuated scattering ratio ceases to be a decreasing function of range. Depending on the SNR in the region below the cloud, this refinement sets cloud base below any readily detectable "leakage" of the signal into the assumed clear region caused by, for example, the transient recovery time of the detectors or by multiple scattering from small particles in dense clouds. Note that this multiple scattering leakage is not pulse stretching (Miller and Stephens, 1999), but instead occurs for instance because the forward scattering angle from the small ice crystals in cold cirrus can be wider than the CALIOP receiver field of view (Reverdy et al., 2015), so that the multiple scattering factor can be slightly larger below the ice clouds than in cloud (Winker et al, 2003). Once all features have been detected in a column, so called "clear air" regions above and below each feature are identified. To initiate a constrained retrieval, the V4 CALIOP extinction algorithm requires a minimum feature-free vertical extent of 2.48 km both above and below a candidate feature. The required effective layer two-way transmittance is then calculated as the ratio of the mean attenuated scattering ratios computed over these below-cloud and above-cloud clear air regions. The fidelity of these estimates relies on the supposition that the backscatter signals in the clear air regions are due solely to air molecules. If this condition is not met (and this is impossible to confirm with absolute certainty), then unless the mean particulate scattering ratios in the two clear air regions are identical, the transmittance measurements will be in error, no matter how small the reported uncertainty, and the constrained retrieval will also be in error. These are bias errors, not random errors, as discussed in Sect. 3b2 of Young et al. (2013), and by del Guasta (1998). Undetected particulate layers above the layer being analyzed can also affect the calculated lidar ratio. Extreme errors can cause the derived lidar ratios to approach and sometimes even exceed the physically acceptable limits of 0.05 sr to 250 sr imposed by the V4 retrieval scheme. Any constrained retrieval (bit 0 set to 1 in the extinction QC flag – See Sect. 2.2.6) in which the derived lidar ratio is equal to either of these limits is to be treated as suspect. When serious errors are encountered in the solution of constrained retrievals, bit 8 is also set, giving an extinction QC flag value of 257.

### 2.2.2 Criteria for determining successful retrievals

The criteria for determining whether a retrieval is successful or not have been modified in V4. The equation for calculating the particulate backscatter at a given range is a transcendental equation that is solved iteratively using a Newton-Raphson algorithm (YV09) while that for the uncertainty is now a simple algebraic expression. In V3 it was solved iteratively. (An improved initialization of the backscatter solution used in V4 is provided in Sect. S2 of the supplementary material.) As shown in Sect. S1 of the supplementary material, the lidar ratio appears in the equations for both the particulate backscatter and its uncertainty at a given range. It is a simple matter to determine if a given lidar ratio is too large for the current input data, because solutions simply do not exist. In V3, retrieved uncertainties in backscatter, extinction and optical depth that grew too large were simply limited to a large value of 99.99. However, in V4, the non-existence of an uncertainty solution or the retrieval of an uncertainty that is excessively large are both considered, along with the non-existence of a backscatter solution, to be indications that the

lidar ratio is too large. These conditions trigger an algorithm that reduces the lidar ratio. Having the number of iterations to achieve convergence of the backscatter solution exceed some prescribed maximum is also an indication that the lidar ratio is too large.

### 2.2.3 New algorithm for retrievals in opaque layers

The sensitivity of extinction and optical depth retrievals to lidar ratio errors increases dramatically as the layer optical depth increases (Y1316). In layers determined to be opaque, even small lidar ratio errors can cause large errors in the retrievals. In V3 and previous versions, the first retrieval attempts were made using the initial lidar ratio prescribed for the layer type and subtype. As it is obviously impossible to select an initial lidar ratio that exactly matches the true value in every actual layer studied; the initial value is virtually certain to be either too large or too small. The former case is easily identified, as it is not

possible to retrieve a complete profile if the lidar ratio is too large.  In these situations, the earlier algorithms reduced the lidar ratio, initially in steps of 1 %, and then in steps of 5 %, until a successful retrieval was obtained (YV09). However, given the extreme sensitivity of the retrieval to the lidar ratio and the relatively coarse reductions, the final values of these reduced lidar ratios may well have been too small. The latter case is more difficult to identify, as successful (although underestimated) retrievals using the initial lidar ratio (QC = 16) can still be obtained even if the initial value is too small.

In order to assess the incidence and likely impact of V3 lidar ratios that were either too high or too low, we examined all opaque ROI clouds detected in the V3 data set during January 2010, with separate analyses done for daytime and nighttime measurements. To estimate the lidar ratio that would be required for a totally attenuating layer, we use a well-validated method initially developed by Platt (1973). Given values for the integrated attenuated particulate backscatter profile through the layer, $\gamma'_p$, the layer effective two-way transmittance, $T_P{}^2$, and the layer multiple scattering factor, $\eta$, the lidar ratio can be estimated

using

$$S_P = \frac{1 - T_P{}^2}{2\eta\gamma'_P} \ .$$                                        (1)

The layer effective two-way transmittance, $T_P{}^2$, is zero for totally attenuating layers, so that Eq. (1) simplifies to $S = 1 / (2\eta\gamma'_p)$. For the V3 analyses, $\eta$ for ice clouds is fixed at 0.6, and $\gamma'_p$ is obtained from the V3 data products.  The $\gamma'_p$ values are corrected for molecular scattering contributions using the steps described in Sect. 3.2.9.1 of Vaughan et al. (2005).  To eliminate artifacts

due to failed surface detections in V3, we require $\gamma'_p$ to be greater than 0.0104 sr$^{-1}$, equivalent to an opaque layer lidar ratio of 80 sr.  Enforcing this restriction eliminated 0.4 % of the initial data set.

For those V3 retrievals where the lidar ratio was not reduced (i.e., QC = 16), the final lidar ratio is, as expected, identical to the initial lidar ratio at 25 sr.  However, the lidar ratio required to attenuate the signal totally (i.e., for $T_P{}^{2\eta} = 0$), computed using Eq. (1), is larger by ~ 24 % at night ($S_{night} = 30.9 \pm 4.4$ sr for 79,049 samples) and by ~ 32 % for daytime measurements ($S_{day}$

$= 32.9 \pm 6.0$ sr for 82,101 samples).  On those occasions when the lidar ratio was reduced (QC = 18; 19,900 nighttime samples

and 21,098 daytime samples), the mean final lidar ratios in the V3 data products were $21.1 \pm 3.0$ sr at night and $20.8 \pm 3.3$ during the day. For these same cases, the mean values of the 'totally attenuating' lidar ratios computed using Eq. 1 are larger by ~3% at night ($S_{night} = 21.7 \pm 3.0$ sr) and ~5 % during the day ($S_{day} = 21.8 \pm 3.6$ sr). The lower lidar ratios reported in the V3 data products for the QC = 18 solutions are most likely caused by the larger step sizes used to reduce the lidar ratio estimate in the V3 extinction retrieval. While these QC = 18 lidar ratio differences are small, the errors introduced in subsequent calculations can be quite large. For example, following the error sensitivity analyses in Y1316, and assuming an optical depth of 3.5, an underestimate of 3 % in the lidar ratio is expected to produce a retrieved optical depth that is, on average, 50 % too low.

From this study we draw two conclusions. First, the initial lidar ratio used for opaque ice clouds in the V3 and earlier data products was, in general, far too low. As a direct consequence, the retrieved extinction coefficients and the estimate of optical penetration into the clouds (i.e., the effective optical depth) were also significantly underestimated. Second, the magnitudes of the lidar ratio reductions were typically too coarse, resulting in an underestimate of lidar ratio even in those cases where the lidar ratio was reduced. Both of these deficiencies have been addressed in the V4 algorithm. The initial value of the lidar ratio for opaque layers is now obtained using Eq. (1), with the effective two-way transmittance assumed to be zero. The multiple scattering factor is derived according to the type of layer being analyzed. For ice clouds, an initial temperature-dependent multiple scattering factor is computed based on the same attenuated backscatter centroid temperature as for semi-transparent clouds (Garnier et al., 2015). For water clouds, the multiple scattering factor is derived from the layer-integrated volume depolarization ratio using the method of Hu (2007). For aerosols, multiple scattering is assumed to be negligible, and hence $\eta$ is fixed at 1.0. However, this assumption may not be valid for opaque aerosol layers, and must be considered especially tenuous for opaque dust layers (Wandinger et al., 2010; Liu et al., 2011). In these cases, the $\eta = 1$ assumption introduces bias errors similar to those incurred by the incorrect specification of lidar ratio (e.g. Y1316). Fortunately, because opaque aerosol layers occur only infrequently (~1 % of all unique aerosol layers (and ~0.2 % of all unique layers) detected in 2012 were identified as being opaque), the effects of these bias errors are somewhat mitigated. To increase the accuracy of the initial lidar ratios computed using Eq. (1), these values are further refined using an iterated version of Eq. (15) in Fernald et al. (1972). (Mathematical details are provided in Sect. S3 of the supplementary material.) Additionally, V4 reductions in the lidar ratio (see Sect. 2.2.4) no longer used fixed, predefined increments, but instead are made inversely proportional to the average retrieved extinction in the layer, so as to make finer adjustments in those cases where the retrieval is more sensitive to lidar ratio errors.

For ice clouds, determining the lidar ratio is a two-step process. In the first step, the multiple scattering factor and an initial lidar ratio are calculated based on the temperature at the centroid of the attenuated backscatter profile (Sect. 2.1.2.1). This lidar ratio is recorded in the V4 data products as the initial lidar ratio. After a successful retrieval of the particulate backscatter and extinction over the whole detected depth of the layer, the multiple scattering factor is recalculated at the altitude of the retrieved particulate backscatter centroid. This recalculation is not necessary for semi-transparent layers, where the centroid altitudes of

the attenuated backscatter profiles provide reliable approximations of the centroid altitudes of the particulate backscatter profiles. For opaque layers, however, the differences between the two can be large (up to a kilometer or more) and hence the multiple scattering factors at the two altitudes/temperatures can be quite different. Since the recalculated centroid height will be at a lower altitude (and likely a warmer temperature) than that of the original attenuated backscatter centroid, the updated

multiple scattering factor is likely to be lower than the initial estimate (see Fig. 1a). An updated extinction solution is then calculated using the updated multiple scattering factor and the newly derived estimate of lidar ratio. The optimized lidar ratio obtained from this second solution is recorded in the data products as the final lidar ratio. One consequence of the refinements made to the multiple scattering factor in this two-step solution process is that the final lidar ratio reported in the V4 data products for opaque layers can sometimes be larger than the initial lidar ratio.

It should be noted that the V4 surface detection algorithm (Vaughan et al., 2018b) has led to large improvements in the assessment of what is and what is not an opaque layer and, as a consequence, when the opaque-layer algorithm can be used. Nevertheless, the accuracy of the new algorithm can still sometimes be compromised by the low SNR conditions that can occur, for example, during daytime measurements of bright, opaque clouds. Low SNR may lead both to missed detections by the surface detection scheme and, within the layer detection algorithm, to missed detections of weaker backscatter signals near

the top and apparent base of the opaque layer. The former problem causes the layer transmittance to be set incorrectly to zero and the latter causes the calculated layer integrated attenuated backscatter to be too small. Both of these errors lead to an overestimate of the initial lidar ratio. In such cases the retrieved extinction profile and optical depth will likewise both be overestimated.

In Fig. 2, we demonstrate the effect, using simulated data, of the different V3 and V4 algorithms on the retrieved extinction

profiles. The top of the simulated cloud layer is at 10 km while the base is at 4 km, although it can be seen in Fig. 2a that the signal is totally attenuated at an altitude of about 7.5 km. The V3 retrieval (Fig. 2b), using a fixed lidar ratio of 25 sr and multiple scattering factor of 0.6, can be seen to trend quickly to zero, significantly underestimating the extinction coefficients at all altitudes. In contrast, the V4 retrieval, which obtains its lidar ratio from the data, stays much closer to the target, trending slightly too low from cloud top to ~8.8 km, then somewhat too high to around 8 km, after which the solution oscillates between

being too high and too low. These excursions from the target result from a combination of small differences in the estimated lidar ratio from the target value and from the effects of noise on the signal.

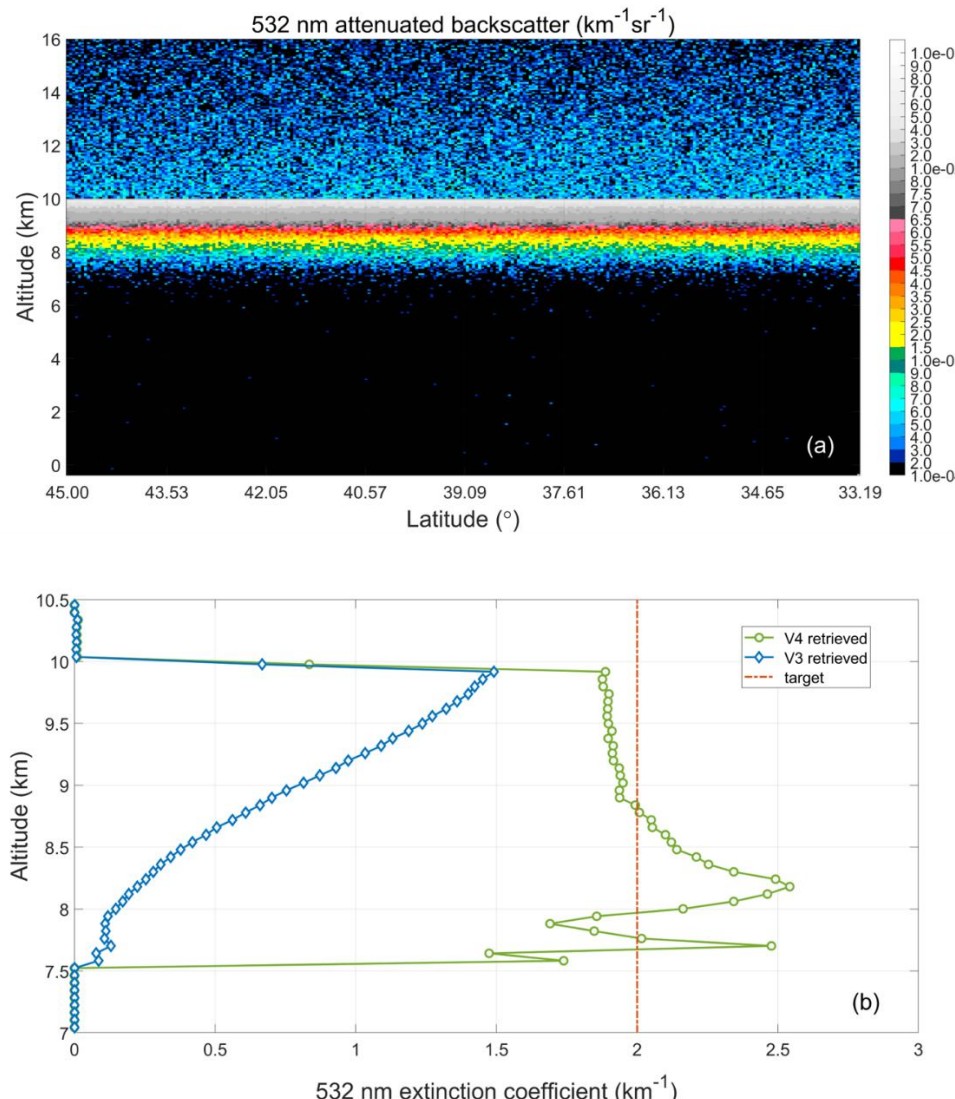

**Figure 2: A comparison of the extinction profiles retrieved by the V4 and V3 algorithms in a simulated totally attenuating ice cloud. The simulated layer has its top at 10 km and its base at 4 km, with a constant extinction coefficient of 2 km⁻¹, yielding an optical depth of 12. The cloud lidar ratio and multiple scattering factor are also uniformly constant values of 33.5 sr and 0.52, respectively. Panel (a) shows a height vs. distance image of the simulated 532 nm attenuated backscatter coefficients. Panel (b) shows the mean extinction coefficients, averaged across the full horizontal extent of the layer, retrieved by the V3 scheme and the V4 scheme. For both versions, individual extinction profiles were retrieved at 5 km intervals prior to averaging.**

### 2.2.4 Revised algorithms for adjusting lidar ratios to achieve a successful solution

There are four main potential errors in the inputs supplied to the extinction algorithm. Two of these errors cause the retrieved particulate extinction to be too high, or for there to be no possible solution at all. The other two cause the extinction to be too low, or even negative.

If the initial lidar ratio estimate is too high, or the correction for the attenuation caused by overlying layers is too large (i.e., when the optical depths of overlying layers are overestimated), then the extinction retrieved in the lower layers will be too high, or, in the worst case, not retrievable at all. While retrieval problems in the topmost layers in the atmosphere can usually be attributed to errors in the initial lidar ratios for these layers, as the retrieval process progresses to lower layers that have an increasing number of overlying layers, the retrieval problems are increasingly likely to be at least partially the result of

attenuation correction errors. As there is no completely reliable way of determining which of these error sources causes an unsuccessful retrieval, unsuccessful retrievals are addressed by reducing the lidar ratio in both cases. Sometimes the lidar ratio needs to be reduced to values that are far below what is considered to be a valid range. These cases are considered to be failed retrievals and are terminated with the QC flag being set to indicate the problem. Where V4 retrievals are terminated at an altitude above the detected base of the feature, a fill value of -333 is inserted in the retrieval arrays of particulate backscatter

and extinction, and their uncertainties, for this altitude and all lower altitudes down to the detected layer base.

When a successful retrieval could not be achieved using a given lidar ratio, the V3 extinction algorithm reduced the lidar ratio by repeatedly applying fixed fractional reductions, initially in steps of 1 % then in steps of 5 %, until either a complete solution was derived or some error condition (e.g., too many solution attempts) occurred [YV09]. The V4 algorithm takes a more nuanced approach. V4 adjustments to the lidar ratio in opaque layers are inversely proportional to the average extinction

coefficient retrieved over the path from the top of the layer to the point where the retrieval fails with the current lidar ratio, multiplied by both the retrieved two-way particulate transmittance over the same range and an empirically derived scaling constant. This produces smaller adjustments where the extinction is higher, as is required to avoid overcorrections. Lidar ratio reductions in opaque layers are limited to a maximum of 1 % on each adjustment. For semi-transparent layers, the fractional reduction is 10 % of the relative uncertainty in the initial lidar ratio.

The opposite problem of the retrieved extinction being too small (or even negative) can occur if the initial lidar ratio is too small or the estimated attenuation for overlying layers is too low. As shown in Y1316, when the optical depth of the layer being analyzed is high, these underestimates can introduce significant errors in the extinction retrieval (e.g., as shown in Fig. 2b). However, whereas cases for which the lidar ratio or the overhead attenuation correction is too large are readily identified (because the extinction solution will diverge before reaching the bottom of a layer), we know of no reliable metric that can

unambiguously identify those instances where the lidar ratio or the overhead attenuation correction is too low. Thus, unlike the V3 algorithm, the V4 retrieval does not increase the lidar ratio when negative particulate backscatter coefficients are repeatedly retrieved from positive attenuated backscatter inputs.

As in V3, retrieval uncertainties and retrieval errors in overlying layers propagate into and are amplified by extinction retrievals in underlying layers. While the random errors (uncertainties) reported in the V4 data products correctly account for random uncertaintes in overlying layers, no estimates are provided for possible bias errors that might also be incurred. However, the combined random uncertainty in the optical depths of overlying layers can be used to estimate the bias error introduced by the

accumulated attenuation corrections. Estimates of the total bias error in the lower layer can then be obtained using the various equations and figures in Y1316.

### 2.2.5 Modification of lidar ratio uncertainties

Lidar ratio uncertainties are major contributors to the extinction and optical depth uncertainties reported in the data products (Y1316). The initial lidar ratio uncertainties are assigned by the various lidar ratio determination schemes (i.e., as in Table 1).

There are some circumstances, however, where the initial lidar ratio uncertainty is modified, or even recalculated completely, within the V4 extinction retrieval algorithm. This final value, whether modified or not, is reported in the V4 data products as the final lidar ratio uncertainty and is expressed as an absolute uncertainty in steradians.

Lidar ratio uncertainties are calculated in constrained retrievals from the uncertainties in the measured apparent two-way particulate transmittance of the layer under analysis and the integrated particulate attenuated backscatter of the layer, as in

Vaughan et al. (2005). In V4, this uncertainty may be reduced if the transmittance plus or minus its uncertainty exceeds the range [0, 1] or if the calculated lidar ratio plus or minus its uncertainty exceeds the range of acceptable values (currently 0.05 sr to 250 sr).  In opaque features, the lidar ratio uncertainty may also be reduced from the default value using the range of lidar ratios calculated from estimates of the minimum and maximum two-way transmittance of the layer, where the minimum is zero (total attenuation) and the maximum is that value that would produce a particulate backscatter at the detected layer base

that is greater than zero where the attenuated backscatter is also greater than zero. If these recalculations produce values that are out of the acceptable range, then the relative uncertainty in the default value is retained.  When lidar ratios need to be reduced in order to retrieve a complete profile over the whole depth of the layer, the reported final lidar ratio uncertainty is likewise reduced so as to maintain the relative uncertainty that is calculated either from the default values or from the modified values just described.

The rules described above do not apply for lidar ratio uncertainties for opaque water clouds because of lidar "pulse stretching". Pulse stretching is caused by the broad forward peak of the scattering phase function of the small spherical droplets typically found in water clouds. The broad forward peak causes an increase in off-axis scattering in water clouds. With the increased multiple scattering found at high optical depths (e.g., in opaque water clouds), this off-axis scattering quickly dominates the backscattered signal with increasing penetration into the layer. Since lidar range information is measured by the time delay of

the backscattered signal, off-axis scattering causes errors in the measured delay, and thereby causes errors in cloud penetration depth measurements. Because of this loss of range information, the particulate backscatter and extinction uncertainties reported for opaque water clouds in the CALIOP V4 data products are now assigned a fill value of -29, indicating zero confidence in

the relationship of the retrieved variables to the related altitude array. We choose, however, to continue to report the retrieved backscatter and extinction for data users who may find value in these quantities as a function of time.

### 2.2.6 Changes in the extinction Quality Control (QC) flags

In order to accommodate the modifications described above, and other changes to the algorithms, the extinction QC flags have been modified in V4. The new values and their explanations are shown in Table 3. The relative occurrence of each of the QC bits for both clouds and aerosols is provided in Table 3. It can be seen that while nearly 93 % of the aerosol layers have identical QC flags in V3 and V4, identical QC values are found in only 65 % of cloud layers. This difference results mainly from the much higher occurrence of QC = 0 in retrievals in aerosol layers.

**Table 2: Extinction Quality Control Flags used in version 3 and version 4.**

| Bit | Value | V3 Interpretation | V4 Interpretation |
|:---:|:---:|---|---|
| 0 | 0 | Unconstrained retrieval; initial lidar ratio unchanged during retrieval process. | Unconstrained retrieval; initial lidar ratio unchanged during retrieval process. |
| 0 | 1 | Constrained retrieval. Solution is constrained by measured two-way transmittance. | Constrained retrieval. Solution is constrained by measured two-way transmittance. |
| 1 | 2 | Initial lidar ratio reduced to prevent divergence of solution to infinity. | Initial lidar ratio reduced to achieve successful solutions for backscatter coefficients and uncertainties throughout the full vertical extent of the layer |
| 2 | 4 | Initial lidar ratio increased to reduce the number of negative backscatter coefficients retrieved for non-negative attenuated backscatter measurements. | Suspicious retrieval (layer or overlying integrated attenuated backscatter too high or lidar ratio reduction excessive) |
| 3 | 8 | Retrieved backscatter exceeds maximum allowable value. Solution terminated. | Lidar ratio has been reduced and has converged but backscatter uncertainty solution still does not exist. |
| 4 | 16 | Feature has been identified by feature finder as being totally attenuating (opaque). | Feature has been identified by feature finder as being totally attenuating (opaque). |
| 5 | 32 | Estimated optical depth uncertainty exceeds maximum allowable value. | Lidar ratio has converged but constrained retrieval still not achieved. |
| 6 | 64 | Lidar ratio has converged but solution still diverging high or low. | Feature has a negative signal anomaly. (Tackett et al., 2018) |
| 7 | 128 | Retrieval terminated at maximum allowable iterations. | Maximum number of attempts at constrained retrieval exceeded. |
| 8 | 256 | No solution possible within acceptable lidar ratio bounds. | No valid solutions exist within acceptable lidar ratio bounds. |

| Bit | Value | V3 Interpretation | V4 Interpretation |
|---|---|---|---|
| 9 | 512 | Unused | Denominator (transmittance) of lidar ratio adjustment factor converged but constrained retrieval still not achieved. |
| 10 | 1024 | Unused | Backscatter solution not achieved but maximum lidar ratio adjustments reached. |
| 11 | 2048 | Unused | Backscatter uncertainty solution not achieved but maximum lidar ratio adjustments reached. |
| 12 | 4096 | Unused | Lidar ratio has been reduced and has converged but backscatter solution still not achieved. |
| 13 | 8192 | Unused | Unused. |
| 14 | 16384 | Unused | Complex feature retrieval failure (see YV09) |
| 15 | 32768 | Fill value or no solution attempted | Fill value or no solution attempted |

**Table 3: A comparison of the V4 and V3 extinction QC flag values for identical layers detected during the years 2007 – 2010. All QC flag values not shown separately are grouped under QC = 99 and indicate the proportion of unsuccessful retrievals. (a) Clouds (b) Aerosols.**

| (a) | Clouds | Version 4 | | | | | | | |
|---|---|---|---|---|---|---|---|---|---|
| | QC flag | 0 | 1 | 2 | 6 | 16 | 18 | 99 | *V3 %* |
| Version 3 | 0 | **32.63 %** | 8.69 % | 0.55 % | 1.05 % | 0.00 % | 0.12 % | 0.10 % | *43.14 %* |
| | 1 | 0.25 % | **1.48 %** | 0.00 % | 0.02 % | 0.00 % | 0.00 % | 0.00 % | *1.76 %* |
| | 2 | 0.25 % | 0.09 % | **1.84 %** | 0.76 % | 0.00 % | 0.15 % | 0.06 % | *3.16 %* |
| | 6 | 0.00 % | 0.00 % | 0.00 % | **0.00 %** | 0.00 % | 0.00 % | 0.00 % | *0.00 %* |
| | 16 | 1.34 % | 0.20 % | 0.29 % | 0.58 % | **3.36 %** | 11.21 % | 0.34 % | *17.33 %* |
| | 18 | 0.09 % | 0.05 % | 2.66 % | 2.58 % | 0.65 % | **26.07 %** | 2.50 % | *34.60 %* |
| | 99 | 0.00 % | 0.00 % | 0.00 % | 0.00 % | 0.00 % | 0.01 % | **0.00 %** | *0.02 %* |
| | *V4 %* | *34.57 %* | *10.51 %* | *5.35 %* | *4.99 %* | *4.01 %* | *37.56 %* | *3.00 %* | **65.38 %** |
| | | | | | | | | | |
| | samples | 1654974 | | | | | | | |

| (b) | Aerosols | Version 4 | | | | | | |
|---|---|---|---|---|---|---|---|---|
| | QC flag | **0** | **1** | **2** | **16** | **18** | **99** | *V3 %* |
| | **0** | **91.70 %** | 0.80 % | 0.38 % | 0.00 % | 0.02 % | 2.42 % | *95.32 %* |
| | **1** | 0.00 % | **0.00 %** | 0.00 % | 0.00 % | 0.00 % | 0.00 % | *0.00 %* |
| **Version 3** | **2** | 0.37 % | 0.00 % | **0.15 %** | 0.00 % | 0.00 % | 0.59 % | *1.12 %* |
| | **16** | 0.85 % | 0.00 % | 0.04 % | **0.39 %** | 1.20 % | 0.37 % | *2.85 %* |
| | **18** | 0.04 % | 0.00 % | 0.01 % | 0.20 % | **0.34 %** | 0.11 % | *0.70 %* |
| | **99** | 0.00 % | 0.00 % | 0.00 % | 0.00 % | 0.00 % | **0.00 %** | *0.00 %* |
| | *V4 %* | *92.97 %* | *0.80 %* | *0.58 %* | *0.60 %* | *1.56 %* | *3.49 %* | **92.59 %** |
| | | | | | | | | |
| | **samples** | **170244** | | | | | | |

## 3 Results of changes

We now present the results of the changes described in Sect. 2, beginning with a study of extinction retrievals in tropical clouds
in Sect. 3.1. In Sect. 3.2 we present an analysis of lidar ratios retrieved in ice and water clouds using the new opaque layer
algorithm (discussed in Sect. 2.2.3). Sect. 3.3.1 compares the optical depths retrieved using these lidar ratios with the V3
values, which were obtained using fixed values of the multiple scattering factors and initial lidar ratios. Optical depths retrieved
in semi-transparent ice clouds using the V4 and V3 algorithms are compared in Sect. 3.3.2, where we demonstrate the
superiority of the V4 results by comparing with MODIS C6 data. In Sect. 3.4 we compare V4 and V3 profiles of median
extinction coefficient as a function of temperature and extinction QC. We end with comments and advice on the use of
extinction QC values for filtering of data in Sect. 3.5 and provide some caveats on using V4 data in Sect. 3.6.

### 3.1 Example: retrievals in tropical clouds

In Fig. 3, Fig. 4, and Fig. 5, we compare the V3 and V4 532 nm analyses of a complex cloud system composed of opaque ice
and deep convective clouds observed over Panama and the adjacent seas by CALIOP on April 24, 2010 around 07:18 UTC.
The example highlights the impact of the new opaque-layer extinction retrieval algorithm. Figure 3 concentrates on the
retrieved backscatter and extinction coefficients, whereas Fig. 4 is mainly concerned with the uncertainties. The level 1 data
were extracted from the files CAL_LID_L1-Standard-V4-10.2010-04-24T07-04-53ZN.hdf and CAL_LID_L1-ValStage1-V3-
01.2010-04-24T07-04-53ZN.hdf. The level 2 data were extracted from the 5-km cloud profile files CAL_LID_L2_05kmCPro-
Standard-V4-10.2010-04-24T07-04-53ZN.hdf and CAL_LID_L2_05kmCPro-Prov-V3-01.2010-04-24T07-04-53ZN.hdf, and

the corresponding vertical feature mask files CAL_LID_L2_VFM-Standard-V4-10.2010-04-24T07-04-53ZN.hdf and CAL_LID_L2_VFM-ValStage1-V3-01.2010-04-24T07-04-53ZN.hdf.

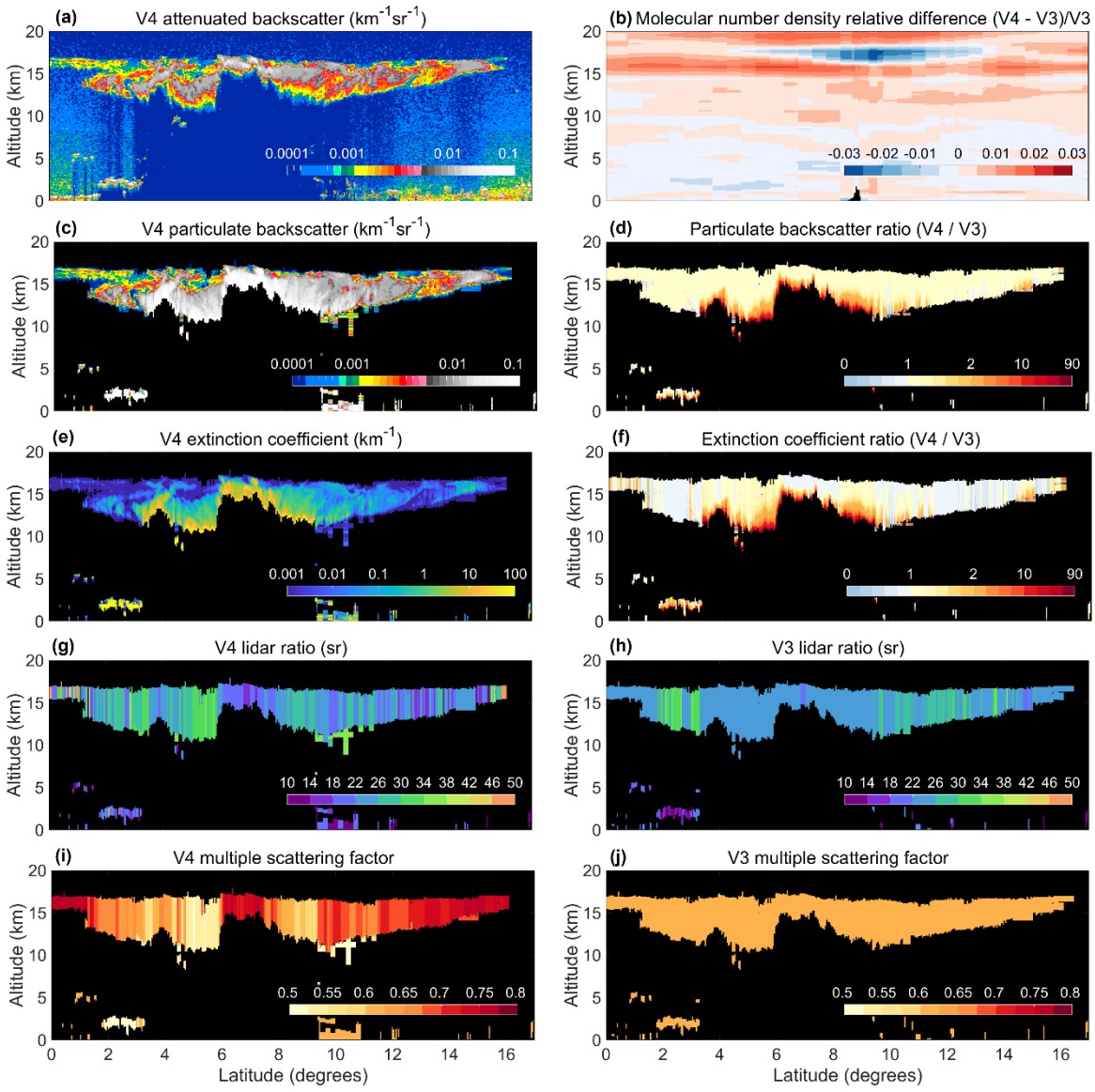

**Figure 3: A comparison of CALIOP V4 and V3 532 nm analyses of dense clouds observed over Central America and surrounding seas around 07:18 UTC on 24 April 2010. See text for details.**

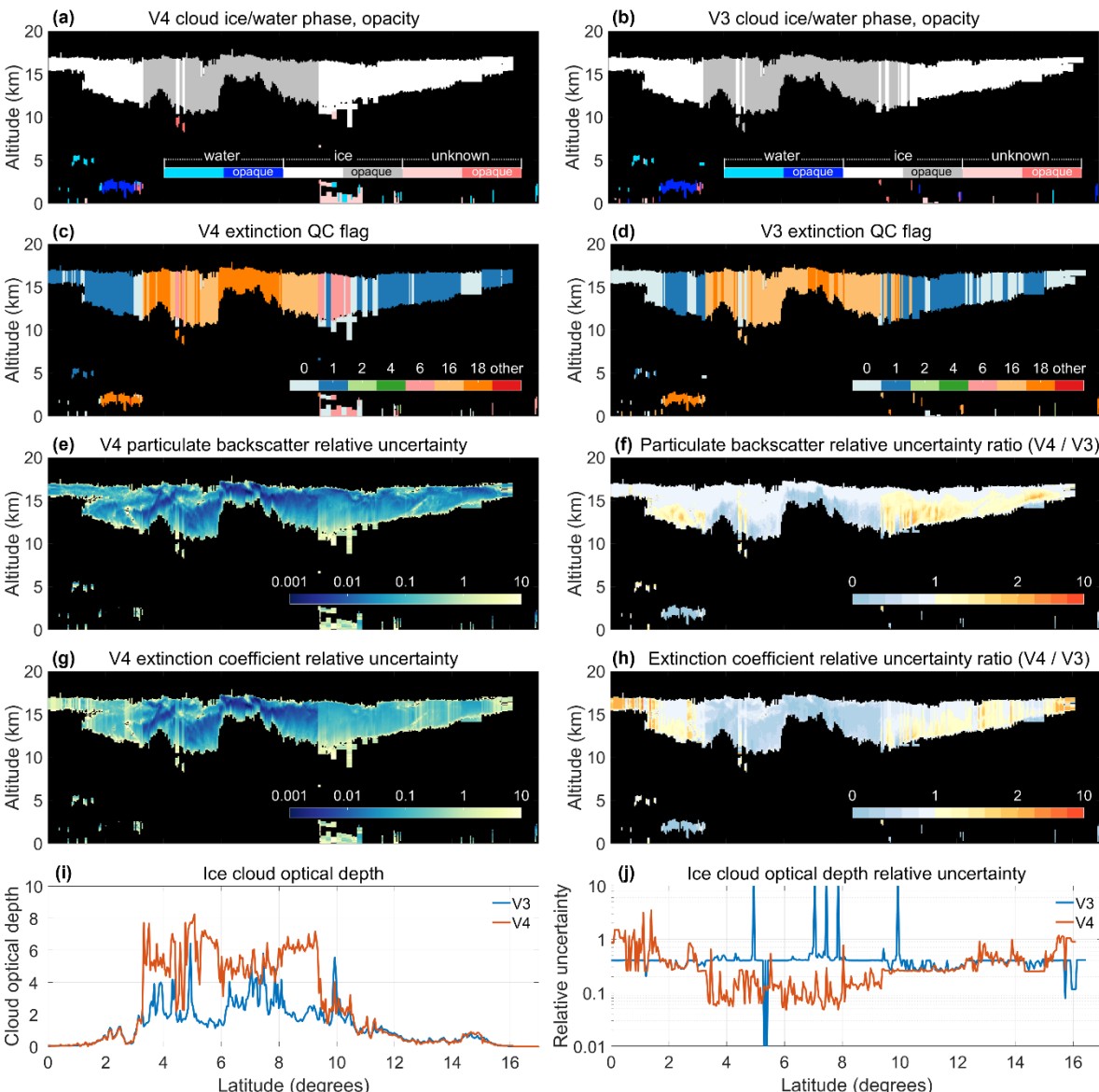

**Figure 4: A comparison of CALIOP V4 and V3 532 nm analyses of dense clouds observed over Central America and surrounding seas around 07:18 UTC on 24 April 2010. See text for details.**

The V4 total attenuated backscatter coefficients (scientific data set (SDS) name = Total_Attenuated_Backscatter_532), plotted in Fig. 3a, differ from V3 because of changes to the calibration coefficients. The V3 and V4 calibration coefficients (Calibration_Constant_532), and their ratios, at the latitudes of interest are shown in Fig. 5a and Fig. 5b respectively. Note that the attenuated backscatter is inversely proportional to the calibration coefficient, so Fig. 5b shows that the V4 attenuated backscatter is between 2 % and 6 % larger than the V3 product over the latitude range shown. There are also differences in the

V4 and V3 molecular number density profiles (Molecular_Number_Density) due to the change of meteorological model used in the analyses.  While both versions use model data produced by the Global Modeling and Assimilation Office (GMAO), V3 used several successive editions of the Goddard Earth Observing System Data Assimilation System (GEOS-5) near real time products, whereas the V4 analyses all use the Modern-Era Retrospective analysis for Research and Applications, Version 2 (MERRA-2) product. The largest changes in the V3-to-V4 calibration ratio occur in the altitude band containing the main ice cloud layer where the V4 molecular number density varies between about 3 % lower to nearly 2 % higher than the V3 density (Fig. 3b). (Because the density differences between V3 and V4 vary with altitude, they are not cancelled by the different V3 and V4 calibration coefficients as these are calculated over a smaller and higher range of altitudes.)  As the molecular backscatter is directly proportional to the number density, and the molecular backscatter is subtracted from the retrieved total backscatter (molecular plus particulate) to give the particulate backscatter, the changes in molecular number density result directly in changes to the retrieved profiles of particulate backscatter and extinction. Note that the changes to the ozone density profiles (not shown) can also cause differences in the retrieved products, though these are typically quite small.

The retrieved particulate backscatter (Total_Backscatter_Coefficient_532) and extinction coefficients (Extinction_Coefficient_532) are plotted in Fig. 3c and Fig. 3e respectively. (Note that, in these products, "Total" refers to the sum of both polarization components of the particulate backscatter, not to the sum of the particulate and molecular components.) Figure 3d and Fig. 3f show, respectively, the ratios (V4 divided by V3) of particulate backscatter and extinction coefficients. The differences in the retrieved quantities result not only from the different calibration (Fig. 5a and Fig. 5b) and different molecular density data, but also from the different lidar ratios, plotted in Fig. 3g and Fig. 3h, and multiple scattering factors, plotted in Fig. 3i and Fig. 3j. The V4 lidar ratios differ from the V3 values where the new algorithms have determined that the cloud is opaque, as indicated by the grey colors in the cloud type plots in Fig. 4a and Fig. 4b, and extinction QC values (Extinction_QC_Flag_532) of 16 or 18 in the extinction QC plots in Fig. 4c and Fig. 4d.  Where the clouds are not opaque, the new V4 lidar ratios and multiple scattering factors were obtained as described in Sect. 2.1.2 while the previous initial values were used for V3. The difference between the multiple scattering factors in Fig. 3i and Fig. 3j is striking, because a constant value of 0.6 is used for all clouds in V3, whereas the V4 values are dependent on the temperature at the cloud attenuated backscatter centroid. In general, lower and deeper clouds have higher backscatter centroid temperatures and these are associated with lower multiple scattering factors. Conversely, the default V4 lidar ratios are generally lower in the higher and thinner clouds.

Between the latitudes of ~3° N and ~10° N, where the upper cloud deck is almost continuously opaque, the V4-to-V3 ratios of backscatter (Fig. 3d) and extinction (Fig. 3f) increase markedly with penetration depth into the cloud and reach factors of 10 or more near the apparent bases of the cloud. This increase with range results from a combination of the smaller V4 calibration coefficient (Fig. 5a and Fig. 5b), a difference that exceeds 4 % between ~ 5° and ~ 9° N, and the higher V4 effective lidar ratios (Fig. 5c), especially in the latitude ranges ~4° to ~6° N and ~8° to ~10° N, where the lidar ratio differences are in the range +5 % to +20 % and the increase in backscatter and extinction with range is especially noticeable. Fig. 2b gives some

insight into how best to interpret these large differences. The V4 effective lidar ratios are higher than the V3 values in the latitude ranges 3° N to 6° N and 8° N to 10° N, and, by scaling the particulate backscatter, lead directly to higher V4 extinctions in these regions. Between 6° N and 8° N, the V4 actual lidar ratios are considerably lower than in V3 and this leads directly to V4 extinctions at the tops of the cloud that are slightly less than the V3 values, as indicated by the presence of pale blue/grey colors in that region in Fig. 3f. These lower V4 lidar ratios result from the higher V4 multiple scattering values associated with the lower temperatures (Fig. 1a) at the higher cloud attenuated backscatter centroid altitudes in this latitude range. Note, however, the effective lidar ratios in this region remain essentially identical between V3 and V4 (Fig. 5c).

The implementation of the opaque layer algorithm in V4 results in generally higher effective lidar ratios between latitudes 3° N and 10° N. Because the retrieval is highly sensitive to the initial value of the lidar ratio at such high optical depths, these higher V4 lidar ratios need to be reduced more often than in V3 as indicated by the more frequent occurrence of QC = 18 (dark orange) values in Fig. 4c than in Fig. 4d. As mentioned earlier, however, while the lidar ratios may have been reduced sufficiently to produce successful retrievals, the final lidar ratios may have still been slightly too large, resulting in retrieved backscatters and extinctions that trend increasingly larger with penetration depth and leading to the high values (dark orange and red regions) near the apparent bases of the clouds in Fig. 3d and Fig. 3f. Nevertheless, as shown with simulated data in Fig.2, the V4 retrievals are substantially more accurate than the V3 retrievals in these opaque regions because the effective lidar ratios were calculated from the data as described in Sect. 2.2.3 rather than using a fixed initial value. The higher V4 optical depths retrieved in the opaque regions of the cloud can be seen in Fig. 4i.

The uncertainties in the retrieved data products result mainly from random noise in the attenuated backscatter coefficients (estimated using the noise scale factors reported in the L1 data products; see Hostetler et al., 2006 and Liu et al., 2006), calibration uncertainties, and uncertainties in the lidar ratios. (There is also a small contribution from the uncertainties in the molecular atmosphere model.) V4 calibration uncertainties are documented in Kar et al., 2018, Getzewich et al., 2018, and Vaughan et al., 2018. Lidar ratio uncertainties are discussed earlier in sections 2.1.2 and 2.2.5, and in Kim et al., 2018. As a consequence of these various increases and decreases in relative uncertainties in V4, and the non-linear nature of the lidar equation at high optical depths, there is no simple relationship between the retrieved backscatter and extinction uncertainties in the two data versions. For example, even though the lidar ratio uncertainties have been reduced, the generally higher V4 lidar ratios lead to higher retrieved backscatter and extinction, particularly with increased penetration depth, and these result in higher relative uncertainties. Focusing on the main ice cloud layer in Fig. 4e, f, g and h, in regions where the cloud is not opaque, the V4 uncertainties are often lower than the V3 values at the top of the cloud but commonly reach or exceed the V3 values at the base of the cloud. In the region where the cloud is opaque (latitudes ~ 3.5° N to ~9.5° N), however, the uncertainties are noticeably less in V4.

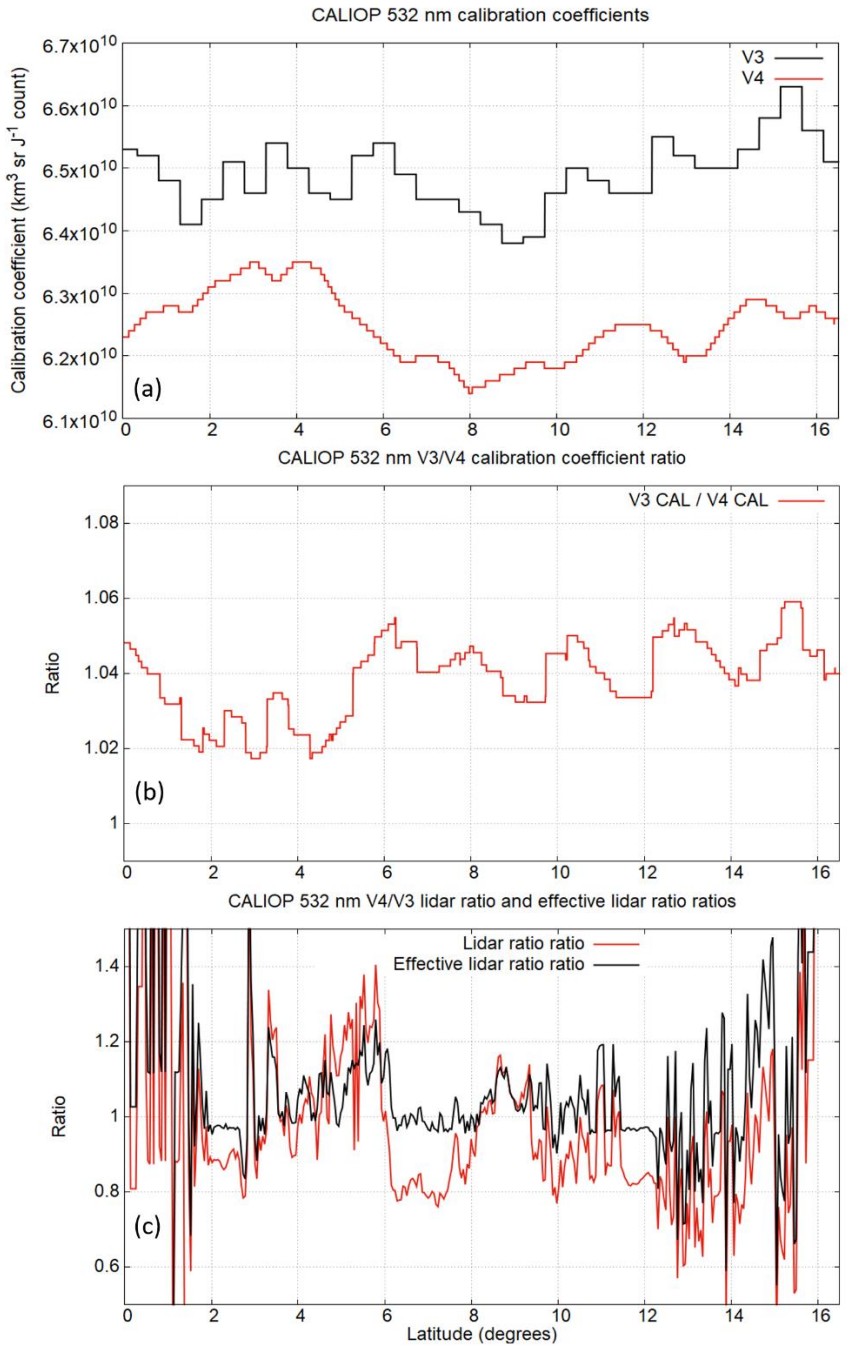

**Figure 5: A comparison of CALIOP V4 and V3 532 nm calibration coefficients and lidar ratios around 07:18 UTC on 24 April 2010. (a) V3 and V4 532 nm calibration coefficients, (b) the ratio of V3 to V4 calibration, (c) the ratios of the V4 to V3 lidar ratios and effective lidar ratios.**

The uncertainties in Fig. 4f and Fig. 4h contribute to the uncertainties in the retrieved cloud optical depths shown in Fig. 4j. These cause higher V3 uncertainties between latitudes ~ 2° N to ~3° N. The considerably lower V4 uncertainty in the opaque regions can be seen between ~ 3° N and ~9° N. The spikes in V3 relative uncertainty exceeding 10.0 result from the limiting of the absolute uncertainty in V3 to 99.99. In V4, the failure of the uncertainty calculation is treated differently, as explained in Sect. 2.2.2.

## 3.2 Opaque cloud lidar ratios

### 3.2.1 Ice clouds

The new opaque layer analysis permits the retrieval of the particulate lidar ratio, a parameter of considerable interest. The accuracy of the lidar ratio depends on the accuracy of the integrated attenuated particulate backscatter of the layer and of the determination of its opacity. The accuracy of the first depends on accurate calibration, accurate determination of the existence of overlying layers and correction for their attenuation, and accurate determination of the full vertical extent of the layer down to the altitude at which the backscatter signal is extinguished (i.e. where the particulate transmittance is truly zero). The accuracy of the second depends on the reliable determination of complete opacity (particulate transmittance is zero) by reliably determining the absence of any surface backscatter signals below the layer. All of these factors are affected by the signal-to-noise ratio, which is lower during the day, so nighttime measurements are expected to be to be more reliable. As an example, we present an analysis of the topmost cloud layers measured during the period 2012 – 2015 and composed of ROI particles. As seen in Fig. 6a, the geometric vertical extent is clearly smaller for daytime data (median = 2.6 ± 1.1 km) than for nighttime data (3.7 ± 1.6 km). The distributions of the integrated attenuated backscatter shown in Fig. 6b are wider during daytime than during nighttime and are slightly skewed towards smaller values. Nevertheless, because the extra geometric extent of the nighttime clouds is at the base, where the signal is highly attenuated, the day and night median integrated attenuated backscatters differ by only 3.3 %. In V3, the initial lidar ratio was 25 sr and the multiple scattering factor was 0.6, which (using Eq. 1) corresponds to a layer integrated attenuated backscatter of 0.033 sr$^{-1}$. The median measured values of the V4 integrated attenuated backscatters differ from this this value by ~6 % (daytime) to ~10 % (nighttime). But as seen in Fig. 6c, the V4 lidar ratios retrieved from these opaque layers are considerably larger than the V3 initial lidar ratio, at 31.5 ± 8.1 sr during the daytime and 31.8 ± 6.7 sr at night. (Recall that the final lidar ratios in V3 were never larger than the initial lidar ratio of 25 sr.)

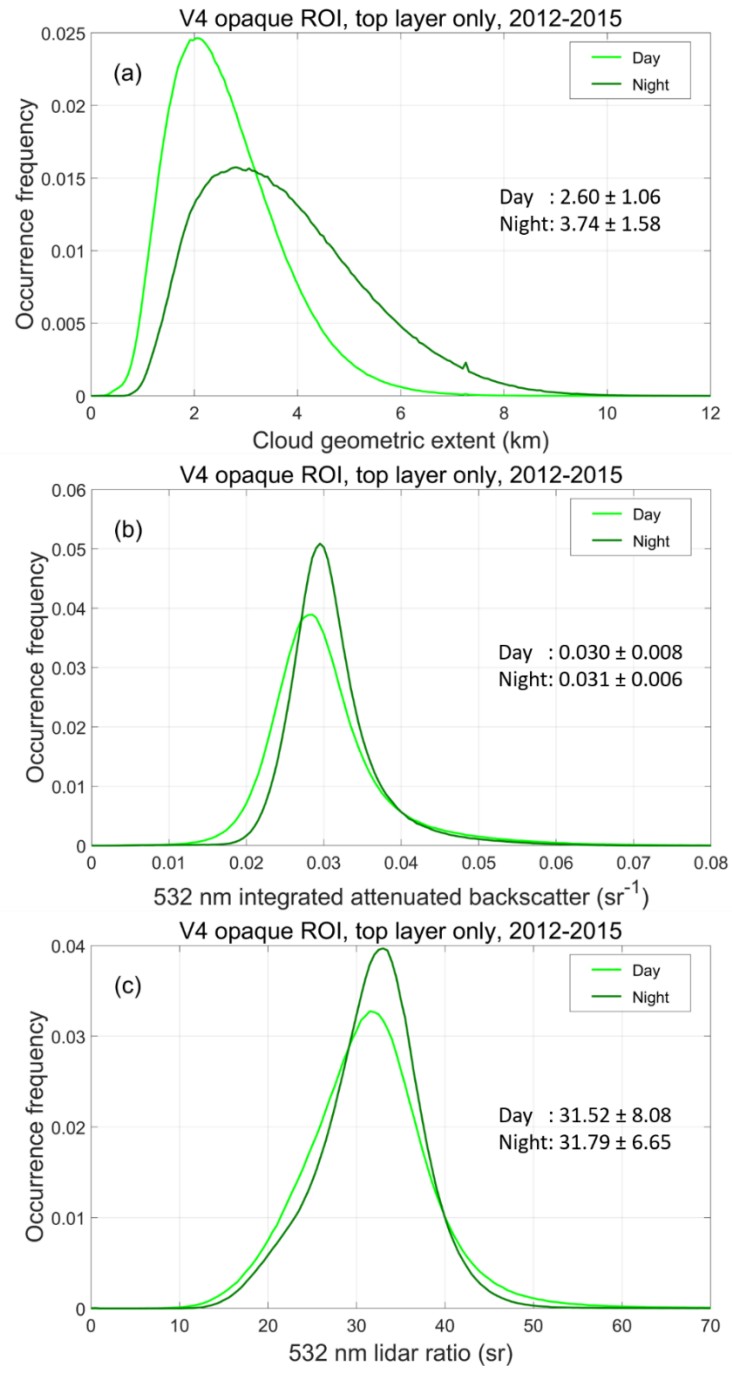

**Figure 6: Histograms of (a) cloud geometric extent, (b) 532 nm integrated attenuated backscatter, and (c) 532 nm lidar ratio for V4 opaque clouds composed of randomly oriented ice (ROI) particles measured during the period 2012 – 2015. Data are for the top layer in the column only. Statistics are median ± median absolute deviation.**

To demonstrate that the new algorithm is indeed producing lidar ratios that are closer to what is expected in opaque clouds than found for V3 retrievals in Sect. 2.2.3, we repeated our analysis of opaque ROI clouds detected in January 2010 using the V4 algorithm. For retrievals in which the lidar ratio was not reduced (QC = 16), we found that the final lidar ratio was within 8 % of the value expected for total attenuation (as calculated using Eq. (1)) for daytime retrievals and only 1.5 % for nighttime retrievals. This is a very marked improvement over the V3 results where the corresponding values were 32 % and 24 %, respectively. For retrievals in which the lidar ratio needed to be reduced in order to achieve a solution (QC = 18), the final V4 lidar ratios were within 4.8 % of the value expected for total attenuation for daytime retrievals and 1.8 % for the nighttime retrievals. The corresponding V3 values were 4.4 % and 2.9 %. It can be seen that the major improvement is achieved in the QC = 16 retrievals where the V3 lidar ratios were markedly too low and would have produced retrieved extinction profiles that were also too low (Fig. 2). This improvement results from the better selection of the initial lidar ratio in V4. A less marked improvement has been achieved in the QC = 18 retrievals. This is because the initial V3 lidar ratio was too high, rather than too low, and was subsequently reduced sufficiently to achieve a solution. The marked improvement in V4 is found in the nighttime retrievals. In the January 2010 sample analyzed, QC = 18 retrievals occurred more than twice as often in the analysis of nighttime data than in the daytime data. This improvement results mainly from the more finely tuned lidar ratio reduction scheme employed in V4.

In Fig. 7 we present maps of the global variations in 532 nm lidar ratios for opaque ROI clouds measured during daytime (Fig. 7a) and nighttime (Fig. 7b) during the years 2012 to 2015. The data are for the same clouds as used to create the histograms in Fig. 6. Given the large differences in daytime and nighttime calibrations, the excellent agreement shown between the distributions of daytime and nighttime lidar ratios is remarkable and gives confidence both in the calibrations and in the new opaque-layer algorithm. The agreement also suggests that the lidar ratio and, by implication, the microphysics in the uppermost regions of opaque ice clouds, do not change significantly between day and night.

A considerable latitudinal variation can be seen in the lidar ratios with the lowest values found over the tropical region and over Antarctica. If the temperature dependence of the lidar ratio for opaque ROI clouds is the same as that for semi-transparent clouds, this behavior could, at least in part, be explained by the variation in lidar ratio with temperature shown in Fig. 1b. Indeed, in the tropics and over Antarctica, the ice clouds occur at altitudes where the temperature is the coolest, as shown in Fig. 7c and Fig. 7d. There are also some land-ocean differences, which suggest that there could be geographic influences on the cloud microphysics. A more detailed analysis, which is beyond the scope of this present work, would be required to determine the causes of the observed distribution.

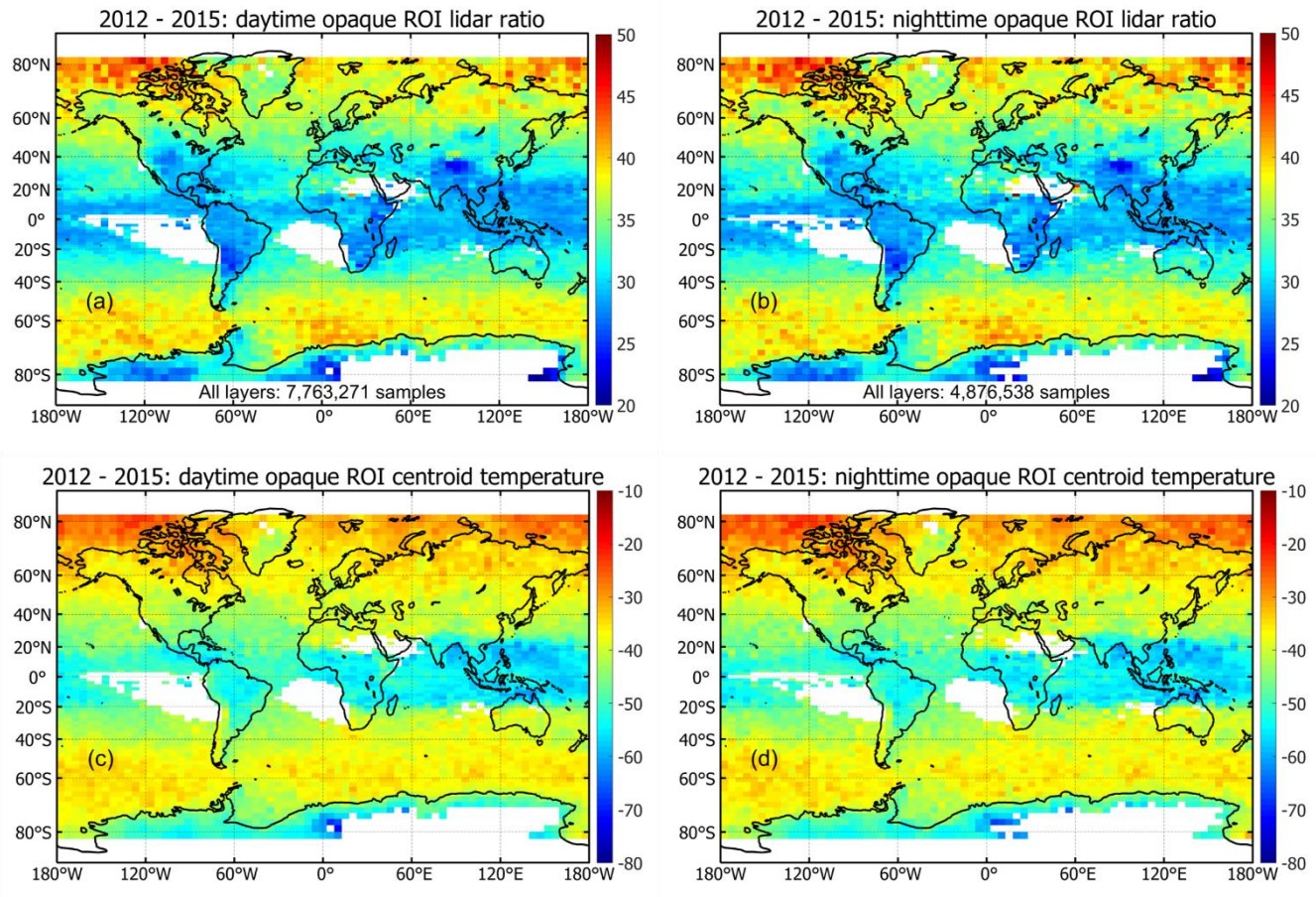

**Figure 7: Global distribution of lidar ratios measured in opaque ROI clouds (a) by day and (b) by night between 2012 and 2015. The corresponding cloud 532 nm attenuated backscatter centroid temperatures are shown in (c) and (d).**

### 3.2.2 Water clouds

5    The new opaque layer algorithms described in Sect. 2.2.3 have also been used to determine the lidar ratio for water clouds. Although retrieval of extinction profiles in opaque water clouds is of low-to-no confidence (as indicated by an uncertainty fill value of -29), the lidar ratio determined from these clouds gives an indication of the possible variation of the lidar ratios that should be used in retrievals in semi-transparent water clouds. In Table 4, we present globally averaged data for the period 2008 through 2010 on the lidar ratio retrieved for opaque water clouds measured during the daytime and nighttime, and on the

10   parameters used in its calculation. Filters have been applied to the data to minimize the effects of undetected overlying layers, to ensure that the layers are composed only of liquid water droplets and that the layers are sufficiently robust to ensure reliable results.

**Table 4: Statistical characterization of global measurements of opaque water clouds during 2008 – 2010, where the clouds were the only layers detected in a 5 km averaged column.  The integrated attenuated backscatter, layer-integrated volume depolarization ratio, and layer-integrated attenuated backscatter color ratio are measured over the detected cloud geometrical thicknesses. The multiple scattering factors are derived using the method of Hu et al. (2007), and the lidar ratios are retrieved using Eq. 1. MAD is the median absolute deviation and N is the number of unique cloud layers used in the calculations. Data have been filtered to minimize the effects of potentially undetected overlying layers. Layer integrated attenuated backscatter is limited to the range (0.021 – 0.111 sr$^{-1}$), layer depolarization to the range (0.03 – 0.39), layer attenuated color ratio to the range (0.90 – 1.50) and overlying attenuated backscatter to values less than 0.02 sr$^{-1}$ and (for nighttime layers only) greater than zero.**

| | Geometric thickness (km) | Integrated attenuated backscatter (sr$^{-1}$) | Depolarization ratio | Attenuated backscatter color ratio | Multiple scattering factor | Lidar ratio (sr) |
|---|---|---|---|---|---|---|
| *Daytime* | N = 7298084 | | | | | |
| **Minimum** | 0.090 | 0.021 | 0.030 | 0.900 | 0.193 | 0.14 |
| **Maximum** | 10.493 | 0.111 | 0.390 | 1.500 | 0.887 | 59.74 |
| **Median** | 0.599 | 0.064 | 0.211 | 1.189 | 0.425 | 18.27 |
| **MAD** | 0.288 | 0.011 | 0.047 | 0.052 | 0.084 | 1.94 |
| **Mean** | 0.747 | 0.065 | 0.211 | 1.189 | 0.436 | 18.54 |
| **Standard deviation** | 0.443 | 0.013 | 0.058 | 0.071 | 0.105 | 2.76 |
| | | | | | | |
| *Nighttime* | N = 3479891 | | | | | |
| **Minimum** | 0.090 | 0.021 | 0.030 | 0.900 | 0.193 | 0.13 |
| **Maximum** | 12.350 | 0.111 | 0.390 | 1.500 | 0.887 | 51.70 |
| **Median** | 0.928 | 0.067 | 0.203 | 1.167 | 0.438 | 16.89 |
| **MAD** | 0.427 | 0.012 | 0.046 | 0.045 | 0.085 | 1.72 |
| **Mean** | 1.152 | 0.068 | 0.204 | 1.165 | 0.448 | 17.26 |
| **Standard deviation** | 0.730 | 0.015 | 0.056 | 0.061 | 0.105 | 2.86 |

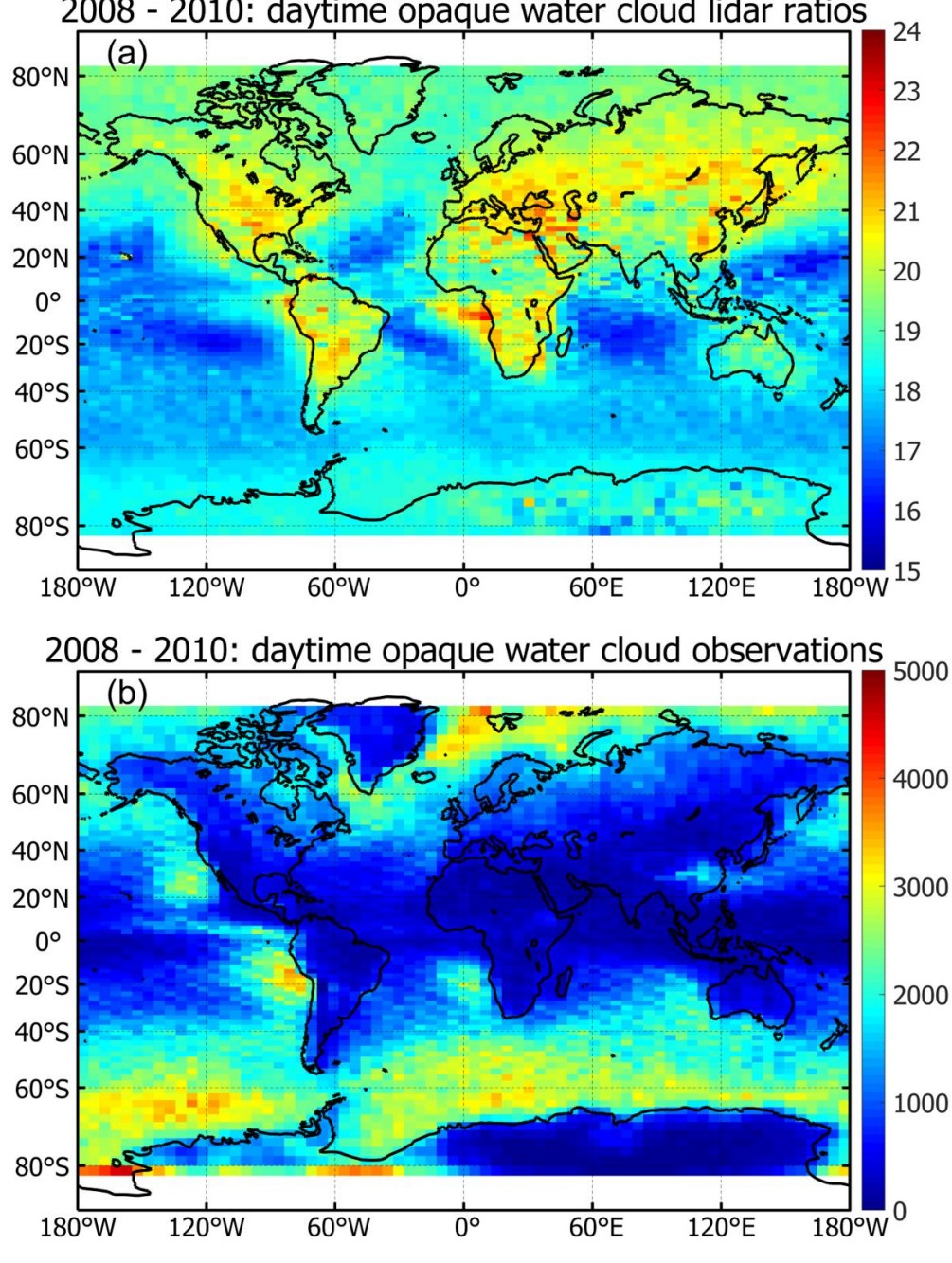

**Figure 8: Mean global distribution of daytime opaque water clouds during 2008 – 2010; (a) 532 nm lidar ratios, (b) sample counts. Data have been filtered as described in the caption for Table 4.**

We note that the mean daytime lidar ratio of $18.5 \pm 2.8$ sr is very close to the theoretically expected value, while the nighttime mean is somewhat lower at $17.2 \pm 2.9$ sr. The low bias in the nighttime lidar ratios is thought to result from a combination of the slow transient recovery of the 532 nm detectors in response to the extremely strong signals from these dense water clouds (Hunt et al., 2009) and a diurnal variation in the instrument's polarization gain ratio (PGR) that is used to calculate the depolarization ratios. The non-ideal detector response is responsible for the larger magnitudes of the nighttime integrated attenuated backscatter. The diurnally varying PGR yields depolarization ratios are slightly, but persistently, lower at night leading to multiple scattering factors that are slightly, but persistently, higher. This combination of high biases in integrated attenuated backscatters and multiple scattering factors at night leads to low biases in the nighttime lidar ratio estimates.

The mean global distribution of the lidar ratios measured in daytime opaque water clouds during 2008 through 2010 is shown in Fig. 8. The quite different global distribution from those of the opaque ice clouds in Fig. 7 is immediately apparent. Generally, higher lidar ratios are seen in the Northern Hemisphere compared with these in the Southern Hemisphere. Higher lidar ratios are seen over the land, and adjacent oceanic regions, particularly in the regions influenced by dust in North Africa and the Middle East, Central Asia and China. Regions influenced by smoke in South America and southern Africa also have higher lidar ratios. While it is tempting to explain the higher lidar ratios in these regions in terms of the Twomey effect (Twomey, 1977), where greater numbers of condensation nuclei produce greater numbers of smaller cloud droplets, or by the increase in extinction through the inclusion of absorbing particles within the cloud water droplets (Mishchenko et al., 2014; Chylek and Hallett, 1992; Miles et al., 2000; Wittbom et al., 2014), a more thorough analysis is required before we can reach these conclusions. Apparent increases in lidar ratio can also be caused by SNR limitations, (particularly in the daytime), which appear as undetected overlying aerosol layers, incomplete determination of the full extent of the cloud layer, or incorrect assignment of opacity because of the missed detection of underlying layers or the surface. While a more thorough analysis is beyond the scope of this current paper, the results presented here suggest that further research could be rewarding.

### 3.3 Cloud optical depths

### 3.3.1 Opaque clouds

We turn now to the optical depths retrieved in opaque clouds using the new methods. Readers are reminded that the CALIOP optical depths reported for opaque clouds are retrieved only over the range from the top of the cloud down to the altitude at which the signal is totally attenuated (i.e., when the signal is indistinguishable from the ambient noise). Consequently, CALIOP opaque cloud optical depth estimates cannot be compared with cloud optical depths measured by passive instruments. In Fig. 9, we present a comparison of the V3 and V4 distributions of optical depths for the topmost opaque (QC=16 and QC = 18) ROI clouds detected during 2012 – 2015. There are several features of note. Most obvious is the marked difference between the V3 and V4 mean optical depths for these opaque clouds. The V3 distributions for day and night peak at optical depths in the range 1.2 to 1.5, or effective optical depths of 0.72 to 0.9. Such low values are insufficient to cause complete

attenuation of the CALIOP signal and reinforce our conclusion that the V3 lidar ratios and, therefore, optical depths, are considerably underestimated.

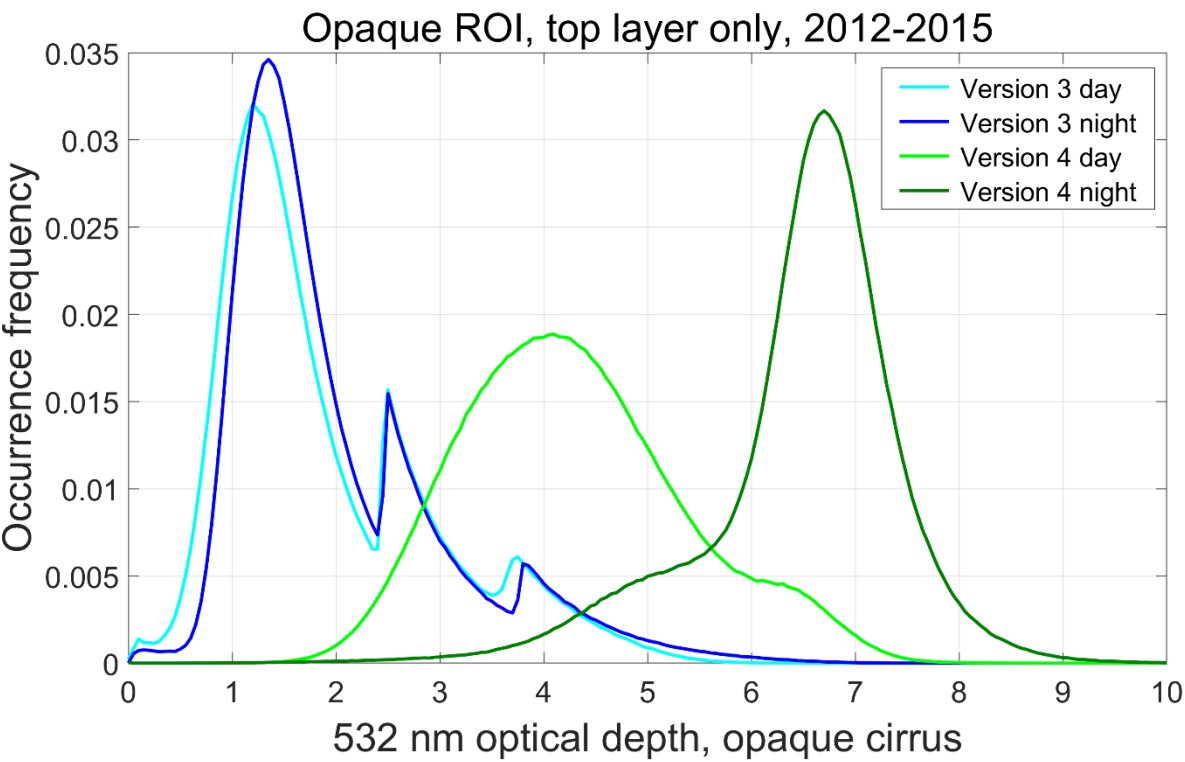

**Figure 9: Distributions of the apparent optical depths measured in opaque ROI clouds during 2012 – 2015 and processed using the V3 and V4 algorithms.**

The second notable feature is the large difference in the daytime and nighttime V4 distributions. This difference is explained by the dramatic shift from QC = 16 to QC = 18 that occurred in the V4 daytime opaque cloud retrievals. The main nighttime peak at ~ 6.6 corresponds to retrievals with QC = 16. Optical depths smaller than about 6 correspond to the far fewer nighttime retrievals in which the lidar ratio required reduction (QC = 18). Conversely, for the daytime data, the small secondary peak at an optical depth of ~ 6.6 corresponds to relative few retrievals in which the lidar ratio has not been reduced (QC = 16), whereas the much larger peak at ~ 4 corresponds to those retrievals where the lidar ratio has been reduced. The much broader daytime QC = 18 peak in V4, when compared with the sharper peaks in V3, results largely from the new lidar ratio reduction scheme with its finer and variable adjustments.

The final feature of note relates to the peaks in the frequency of occurrence of V3 optical depths at around 2.5 and 3.85. When multiplied by a multiple scattering factor of 0.6 that was used for clouds in V3, these values correspond to effective optical depths of 1.5 and 2.5 respectively.  The peaks occur because of the fixed reduction steps applied in the V3 algorithms to the

lidar ratio in the case of a solution that would not converge. The peak at an optical depth of 2.5 corresponds to a lidar ratio reduction of 5 % and that at 3.85 to a reduction of 1 %. These reductions are too coarse in optically dense layers, where the transmittance is close to zero, and cause the artifacts seen in the figure. (See Omar et al., 2009 for a detailed explanation.) These artifacts are not seen in V4 thanks to the refined reduction of the lidar ratios described in Sect. 2.2.4.

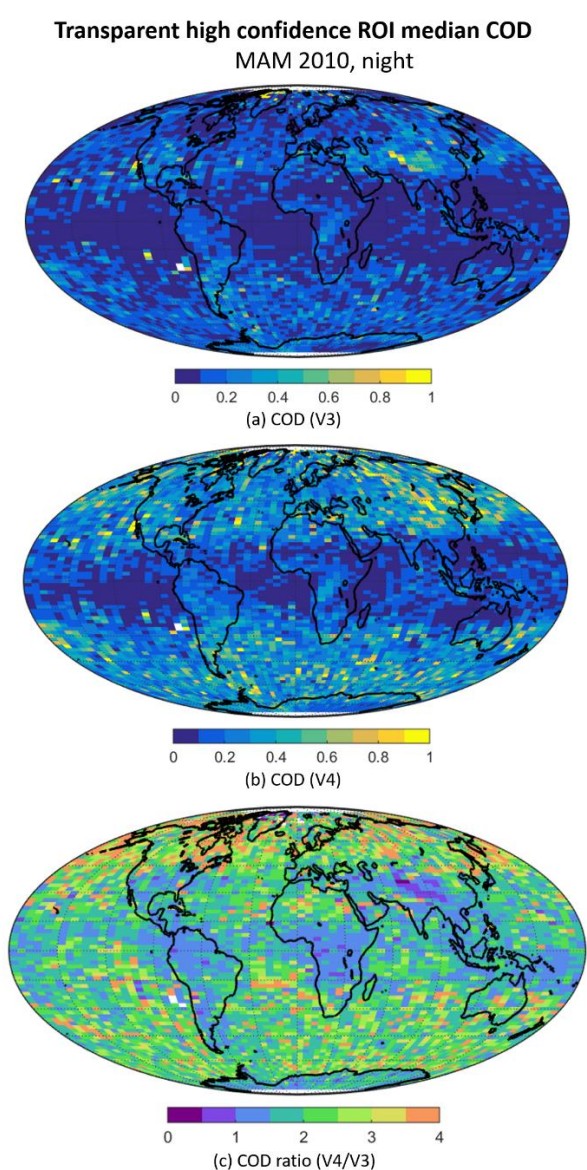

**Figure 10: Median cloud optical depths (CODs) for semi-transparent, high-confidence nighttime ROI clouds for the period March to May 2010 inclusive, nighttime only. The area-weighted average optical depths from the maps are 0.137 for V3 (77,230,457 samples) and 0.266 for V4 (78,808,302 samples).**

### 3.3.2 Semi-transparent clouds

Global maps of the nighttime optical depths for semi-transparent high confidence ROI clouds (QC = 0, 1 or 2) are presented in Fig. 10. For both V3 (Fig. 10a) and V4 (Fig. 10b), the optical depths in the tropical regions, particularly over the oceans, are considerably lower than at higher latitudes. The V4/V3 COD ratios (Fig. 10c) are higher at middle to high latitudes and suggest that the changes between V4 and V3 differ, to some extent, over land and ocean. For example, the lower V4/V3 ratios over South America, Central and Southern Africa, and especially over the Himalayas are rather distinct. In general, the area-weighted average optical depths are considerably higher for V4 (0.266) than for V3 (0.137). The dominant cause of the larger V4 optical depths is the higher lidar ratios (Fig. 1b) used in the V4 retrievals.

As mentioned previously, Holz et al. (2016) showed that while CALIOP V3 constrained (QC = 1) optical depths in semi-transparent ice clouds agreed rather well with MODIS C6 optical depths, the unconstrained retrievals (QC = 0, 2) were substantially smaller than the MODIS values. As a conclusive demonstration of the major improvement in the V4 optical depths, in Fig. 11 we compare MODIS C6 optical depth estimates for semi-transparent ice clouds measured over the ocean during daytime, with collocated CALIOP V3 and V4 optical depths. The MODIS 1 km C6 optical depths were collocated with the centers of the CALIOP 5 km data segments. The samples selected for comparison were limited to those cases for which only one high-confidence ROI cloud was detected in the CALIOP 5 km column and no clouds were detected at single shot resolution and subsequently removed by the boundary layer cloud clearing algorithm. The layers selected for comparison were required to have identical base and top altitudes in both V3 and V4, and they were further restricted to those cases for which the extinction QC flag was either 0 or 1. Data for January 2008 are shown in the upper row in Fig. 11 while data for July 2008 are shown in the lower row. Whereas the CALIOP V3 data presented in Fig. 11a and Fig. 11c are markedly lower that the MODIS C6 values, the comparisons in Fig. 11b and Fig. 11c show an excellent agreement of the V4 optical depths with the MODIS values. The good agreement of the CALIOP V4 and MODIS C6 optical depths gives us confidence in the validity of the new multiple scattering factors and lidar ratios presented in Sect. 2.1.2.

### 3.4 Median ice cloud extinction coefficient profiles

With the changes to the lidar ratios and multiple scattering factors, which, as described in Sect. 2.1, are now functions of temperature, V4 extinction coefficients would be expected to show a rather different height and temperature dependence from that shown by the V3 retrievals. We examine these differences in this section.

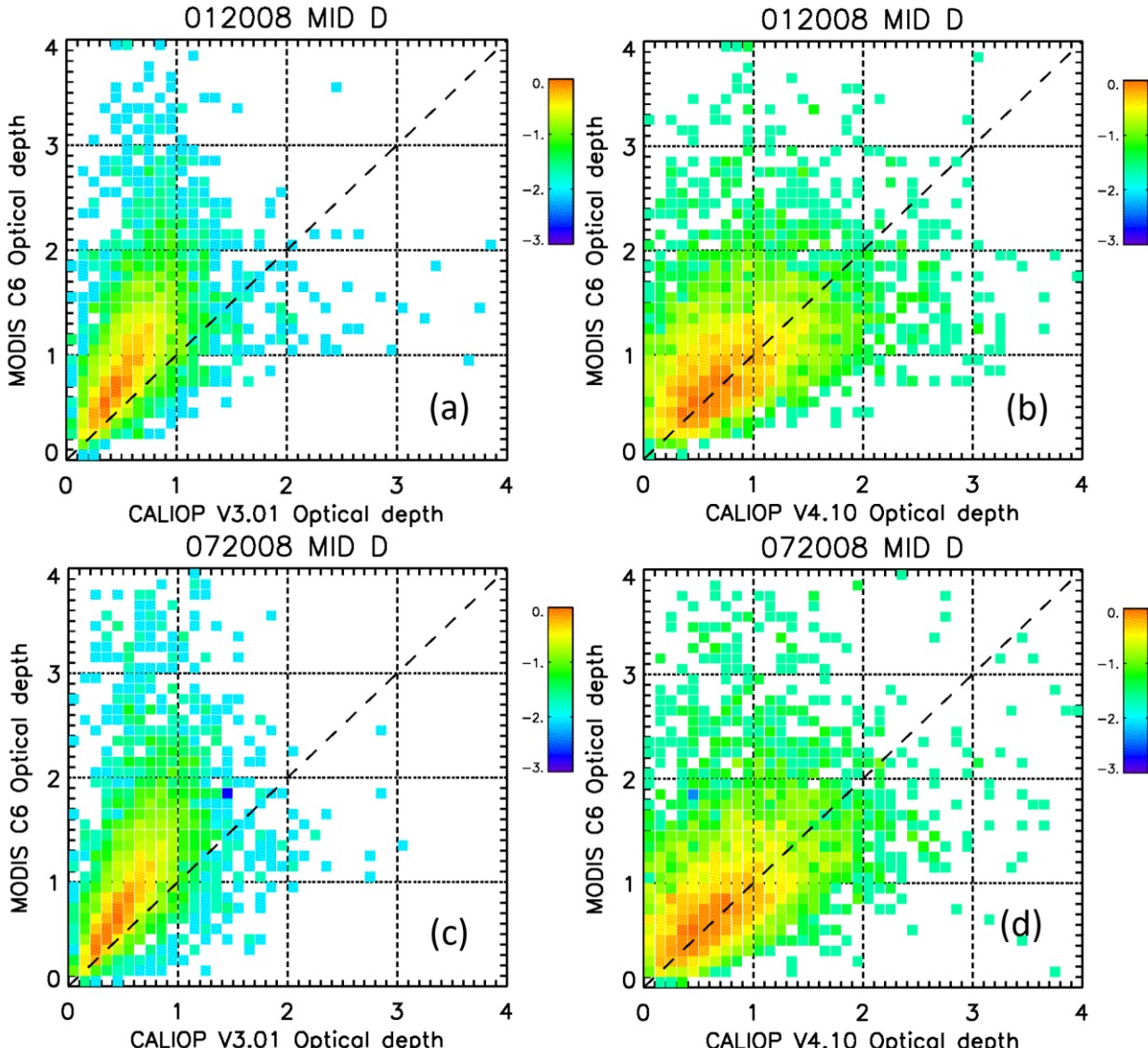

**Figure 11: A comparison of collocated MODIS C6 optical depths with CALIOP optical depths retrieved using the V3 (left-hand panels) and V4 (right-hand panels) algorithms for January 2008 (top row) and July 2008 (bottom row). Data were sampled over oceans in daytime and from both hemispheres between 30° and 60° latitude. Color scale indicates number of samples (on log 10 scale) normalized to the maximum value in each panel. The 1-km MODIS C6 samples were collocated with the center of the CALIOP 5-km samples. Data were obtained from the ICARE CALTRACK product. Samples were restricted to cases where there was only one V4 cloud in the column with no boundary layer clouds cleared at single shot resolution. All V4 clouds were identified as high confidence ROI, whereas the V3 phase was ice (either ROI or HOI) and the MODIS phase was ice. V4 data were restricted to QC = 0 or QC = 1, but all transparent ice clouds were accepted in V3. Samples were further restricted to cases where cloud tops and bases were identical in V4 and V3.**

In Fig. 12, we compare V3 and V4 cloud extinction coefficients plotted as a function of temperature. The profiles are the medians of the topmost layers, identified as clouds with a CAD score of 50 or greater (i.e. classified as clouds with medium confidence or better), composed of ROI crystals and detected between 60° N and 60° S during September 2014. The profiles are grouped according to the extinction QC values and into daytime and nighttime results. The numbers of samples at each temperature are shown in the panels on the left with the median extinctions on the right.

For unconstrained retrievals, in which the lidar ratio is unchanged from the initial value (QC = 0, Fig. 12a, Fig. 12b), the median extinction profiles in V4 are larger at all temperatures above -72 °C and this difference increases with temperature and as would be expected from Fig. 1b, where the V4 lidar ratios are shown to increase relative to the V3 values. There are also fewer QC = 0 retrievals in V4 than in V3 (Fig. 12a), mostly because many retrievals that were assigned QC = 0 in V3 are accepted as constrained retrievals in V4 (see Sect. 2.2.1), so that clouds of small optical depth that were QC = 0 in V3 are QC = 1 in V4. The smaller occurrence of QC = 0 retrievals in V4 and the increase in retrievals with QC = 1 can be seen in Fig. 12a and Fig. 12c respectively. A closer agreement between daytime and nighttime QC = 0 extinction profiles is seen in V4.

The large increase in the number of constrained retrievals (QC = 1) in V4 occurs for all temperature levels for both nighttime and, particularly, daytime retrievals (Fig. 12c). The increase in nighttime retrievals is greater at higher altitudes (lower temperatures) than for the daytime retrievals, possibly because the SNR at the higher altitudes is often too low by day to permit detection of tenuous features or reliable measurement of their effective transmittances. The V4 median extinction coefficients are lower than the V3 values at temperatures below ~ -20 °C. This is explained by the larger fraction of clouds of small optical depths in the population of QC = 1 retrievals in V4 than in V3. The mean optical depth for QC = 1 retrievals in V4 is 0.560 ± 0.599, whereas for V3 the QC = 1 mean is 0.988 ± 0.558. This difference is the direct result of the algorithm changes described in section 2.2.1.

The use of higher lidar ratios in V4 would be expected to lead to a greater number of retrievals in which the lidar ratio needs to be reduced in semi-transparent clouds (QC = 2, Fig. 12e). While this is seen in the nighttime comparisons, the daytime comparison is not as straightforward. Given the use of generally higher ice cloud lidar ratios in V4 than in V3, the almost uniformly lower V4 extinction coefficients in V4 than in V3 for QC = 2 (Fig. 12f) is, at first, surprising. In V4, however, as explained in Section 2.2.2, the lidar ratio is reduced if the calculation of either the backscatter or its uncertainty fails. In V3 it was not reduced if the uncertainty calculation failed and the uncertainty simply limited to 99.99 units. The consequence is that the final, reduced, lidar ratios in V4 are lower than in V3 and this results in lower extinction coefficients. It is important, too, to remember that QC = 2 is a very special case. In V4, layers that are originally assigned QC=2 get additional scrutiny and, if the solutions are found to be unrealistic, bit 2 of the QC flag is set, so that instead of being assigned QC = 2 they are now assigned QC = 6. This additional quality assurance test, which was not implemented in V3, helps to further explain why the V3 extinction coefficients are larger than the V4 values for QC = 2. (This is yet another case of not having the same underlying

cloud populations, even though the QC values are the same. Note, in particular, the huge decrease in the number of daytime V4 retrievals at higher temperatures, which occur at the lower part of the cloud layer (Fig. 12e), and may be a result, in part, of changes to the ice-water phase algorithm (Avery et al., 2018). In V4, some of the erroneous solutions that were left intact in the V3 analyses have now been identified and labeled with the correct QC flag.)

As discussed in earlier sections, the effective lidar ratios used in V3 retrievals in opaque layers were almost always too low. This produced an underestimate of the retrieved extinction, which increases with penetration depth, as can be seen the comparison with the V4 profiles shown in Fig. 12h. Because of the smaller reductions in lidar ratio in response to unsuccessful retrievals in V4, described in Section 2.2.4, the final V4 lidar ratios will also be larger in QC = 18 retrievals, again leading to generally higher retrieved extinctions (Fig. 12j). The close agreement between the V4 daytime and nighttime QC = 16 profiles
(Fig. 12h), despite large diurnal variations in the calibration coefficients, gives confidence in the new algorithm and also in the general performance of the new surface detection algorithm, which determines layer opacity. The marked reduction in the number of daytime QC = 16 (Fig. 12g) retrievals and corresponding increase in QC = 18 (Fig. 12i) retrievals result both from these factors and from the general increased likelihood of failures in the initial retrievals when using higher lidar ratios, particularly at high optical depths. While there has been a ~60 % reduction in the total (day plus night) QC = 16 retrievals in
V4, there has been a ~ 100 % increase in the total number of QC = 18 retrievals. The total number of successful opaque layer retrievals has increased by only ~6 %.

Finally, a comment must be made about the kink, or increase in slope, in almost all of the V3 and V4 extinction profiles at a temperature of ~ -15°C. It is possible that this results from a change in phase near the bases of clouds. Currently, the same ice water phase is assigned to all levels of a detected layer. A retrieval in a layer determined to be ROI would be assigned an initial
lidar ratio, which would be considerably larger than that for a water cloud, for the whole depth of the layer. While this would produce a satisfactory result in the regions where the particles were truly ROI, in regions where the particles were water droplets or a mixture of water and ice, the ROI lidar ratio would be too large and an overestimate the extinction coefficient would result.

### 3.5 Comments and advice on the use of CALIOP extinction QC values

Extinction QC flags are often used to filter data for quality during analyses, and the question is sometimes asked as to which values indicate retrievals that are "trustworthy" or "reliable". Unfortunately, there is no absolute or straightforward answer to this question and the QC flags should be used while bearing in mind various caveats. That said, only those data with QC flag values of 0, 1, 2, 16 and 18 should be used for reliable scientific analyses.

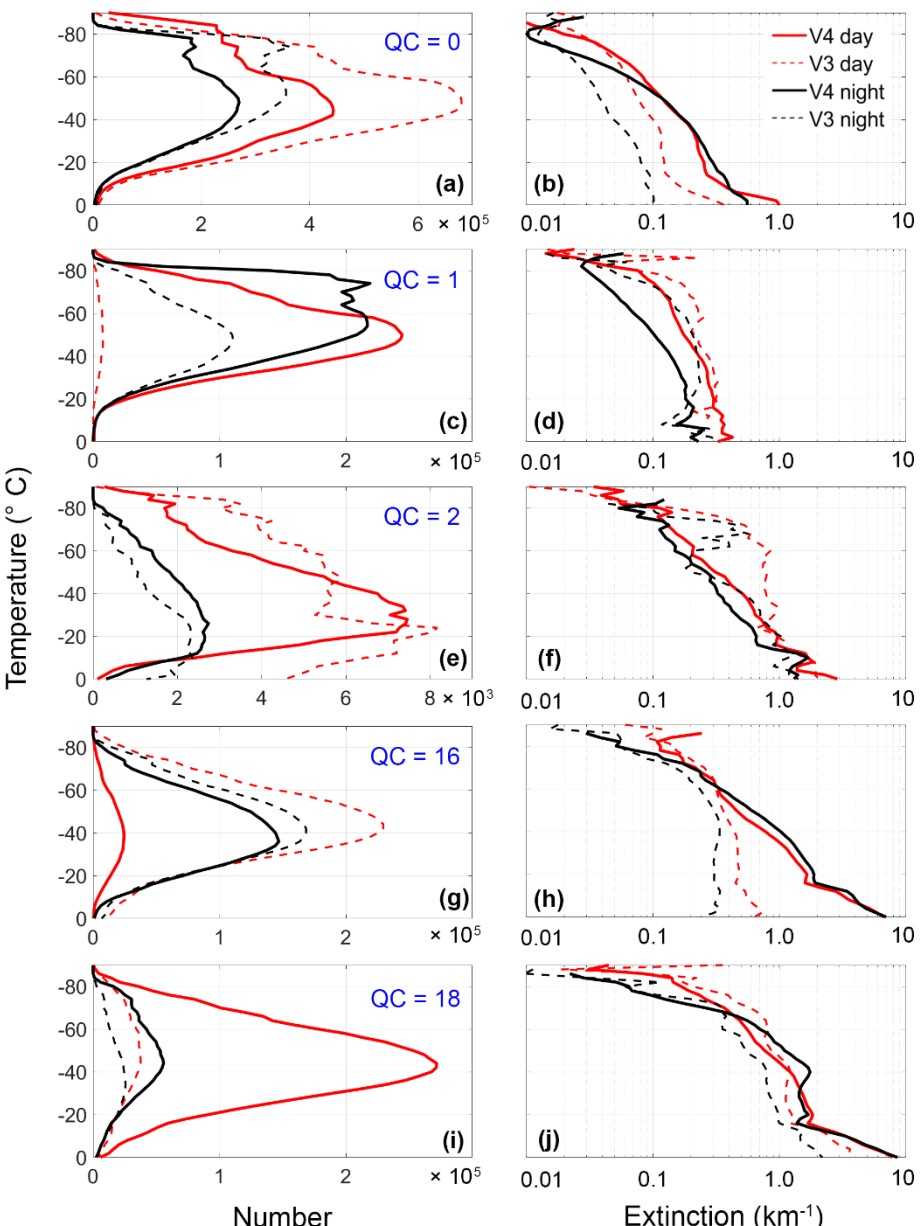

**Figure 12: A comparison of version 4.1 and version 3.3 median ROI cloud extinction profiles measured during September 2014. All clouds were the topmost detected features in the atmospheric column. Daytime and nighttime median extinctions are plotted as functions of temperature and QC value in the column on the right while the corresponding numbers of contributing samples are plotted in the column on the left.**

Constrained retrievals (QC = 1) are generally regarded as the highest quality retrievals. However, even QC = 1 retrievals will sometimes be affected by calibration errors or errors caused by missed detection or incorrect calculation of the attenuation of overlying layers. These situations are considered as normalization errors and their effects described in Y1316. Constrained retrievals for which the final lidar ratio approaches the specified limits (0.05 sr to 250 sr) should be considered suspicious, as

they most likely represent a mismatch (due to noise) between the measurements of integrated attenuated backscatter and apparent two-way transmittance for a given layer. The highest quality constrained retrievals are found in those features that (a) are the topmost layer detected, (b) have relatively low values of both the measured two-way transmittance and its relative uncertainty, and (c) have low values of overlying attenuated backscatter. Constrained retrievals account for ~ 11 % of all V4 cloud solutions and ~ 1 % of all aerosol solutions.

Unconstrained retrievals in semi-transparent layers in which the lidar ratio has not been reduced (QC = 0) usually occur in layers of low particulate extinction and low optical depth. (The higher the optical depth, the higher the sensitivity on the lidar ratio (Y1316).) Because the optical depths are low, so too are the internal two-way transmittance corrections. The result is that the retrieved particulate backscatter profiles are relatively insensitive to errors in the lidar ratio and thus are of good quality in low optical depth layers. The retrieved particulate extinction profile, however, because it is obtained by multiplication of the

retrieved backscatter profile by the lidar ratio, will reflect any errors that exist in the lidar ratio. Unconstrained retrievals account from ~ 41 % of all V4 cloud solutions and ~ 92 % of all aerosol solutions.

As explained in Sect. 2.2.2, unsuccessful retrievals can occur if either the input signal or the lidar ratio is too large, or some combination of both. When this occurs, the lidar ratio is reduced in an attempt to generate a successful retrieval. Successful retrievals of transparent layers for which the lidar ratio has been reduced have QC = 2, and account for ~ 5 % of all V4 clouds

solutions and less than 1 % of all aerosol solutions. Overestimates of the initial lidar ratio are most likely the cause of those QC = 2 layers that are the uppermost layer in a column. For all other layers, overestimates of the optical depths of overlying layers are at least equally likely to result in QC = 2 solutions. As a general rule, layers with extinction QC = 2, are likely to have errors that are larger than in those with QC = 0 or QC = 1. The highest quality QC = 2 retrievals are expected to be topmost layers having low overlying integrated attenuated backscatter.

The initialization of retrievals in opaque layers in V4 is quite different from that in V3, as the initial lidar ratio is no longer a fixed (albeit layer-type dependent) value but is instead calculated from the attenuated backscatter profile. In general, this results in higher initial lidar ratios in V4, which in turn lead to a vastly increased fraction of opaque-layer retrievals in which the initial lidar ratio requires reduction (QC = 18). Because solar background signals degrade the daytime SNR and make accurate layer (and surface) detection more difficult, nighttime retrievals are considered to be of higher quality than daytime retrievals for

both QC = 16 and QC = 18. For similar reasons as used in the comparison of retrievals with QC = 0 and QC = 2, QC = 16 retrievals are likely to be more accurate than those with QC = 18, although it is possible that the lidar ratio is still too high in both cases, if not high enough to require (further) reduction. This would cause the retrieved profiles to become increasingly

overestimated with penetration depth. Because of the improved adjustment of lidar ratios in V4, however, any retrieval errors should be substantially smaller than in V3. QC = 16 retrievals account from ~ 3 % of all V4 cloud solutions, whereas QC = 18 retrievals account for ~ 29 %. QC flags of 16 and 18 combined account for less than 1 % of all V4 aerosol retrievals.

### 3.6 Caveats regarding the retrieval of extinction from elastic backscatter lidars

The accuracy of the CALIOP V4 extinction retrievals has been substantially improved relative to previous versions, primarily through the increased use of constrained retrievals and, for unconstrained retrievals, the use of initial lidar ratios that more accurately characterize the identified cloud and aerosol types. Nevertheless, there remain some potential pitfalls in using the CALIOP extinction data products and some areas in which improvements may be possible. In general, these pitfalls are not specific to the CALIPSO lidar or the V4 retrieval scheme, but instead are applicable to the retrieved data products produced

by any elastic backscatter lidar.

The fundamental caveat relates to the lidar ratios used in unconstrained retrievals. Although the V4 initial values are more representative than in previous versions, natural variability still exists in any aerosol or cloud type. The magnitudes of the uncertainties ascribed to the CALIOP layer types are thus largely driven by our knowledge of type-specific natural variability, so that the downstream uncertainties associated with the optical profiles reported in the CALIOP V4 data products will

generally account for both noise in the measurements and natural variations in layer properties. However, our ability to tightly constrain the bounds of this natural variability in (for example) static regional models is confounded by long range transport. This is especially so for aerosols such as dusts and smokes, which show wide variations in lidar ratio (Schuster et al., 2012; Burton et al., 2012) and are frequently measured in large concentrations far from their source regions (Uno et al., 2009; Yu et al, 2015; Di Biagio et al., 2018). As a result, within any single layer, the degree to which the lidar ratio used in a retrieval

differs from the actual value has a direct impact on the accuracy of the retrieved extinction coefficients and optical depths and this sensitivity increases with the optical depth of the feature (Y1316). For low optical depths, the relative error in the retrieved optical depth is closely approximated by the relative error in the lidar ratio, but for higher optical depths, the optical depth error increases by a factor of approximately $(\exp(\tau) - 1$; see Eq. 40 to 43 in Young et al., 2013). As a consequence, even for a cloud with an optical depth as low as 3, in order to retrieve that optical depth to an accuracy of 10 %, it is necessary to specify

the lidar ratio to an accuracy of 0.1 %. Note, too, that while the estimated uncertainties (section S1.2 in the Supplementary material) also increase with depth of penetration and increasing retrieved optical depth, they may underestimate the true uncertainties when the uncertainties in the input parameters are large and, therefore, do not adequately approximate the retrieval errors (see section S1.3 in the supplementary material).

There are some circumstances in which constrained retrievals may also be in error. As explained in Sect. 2.2.1, particulate

scattering in regions used for normalization can lead to biased results. Also, as discussed by Reverdy et al. (2015), forward scattering from small ice crystals within a cloud can cause an enhancement in the backscatter signal measured below the cloud that decreases with range below cloud base. They further suggest that, for CALIOP signals, the rate of decay is so long that it

is only really notable for cirrus that are composed of small particles (e.g., 10 μm – 20 μm) and have relatively high optical depths. When applying the CALIOP two-way transmittance estimation algorithm, the impact of such an enhancement would be to produce a constraint that is biased high, with the result that both the retrieved optical depth and lidar ratio would be biased low. However, these conditions occur relatively infrequently. For all V4 constrained retrievals of ice cloud profiles measured between 60° S and 60° N during the years 2011 – 2015, only 2.1 % have both centroid temperatures below –70 °C, where particles can be small (Heymsfield et al., 2014), and optical depths greater than 0.5.

If multiple scattering induced biases are suspected, the lidar ratios and optical depths where constrained retrievals are employed should be compared with the same parameters in adjacent columns that use unconstrained retrievals. In any case, constrained retrievals in which lidar ratios and optical depths have high relative uncertainties should also be regarded with caution. Finally, in order to assess the likely impact of these potential errors, we refer the reader to the comparison of CALIOP V4 and MODIS C6 optical depths presented in Section 3.3.2 and in Fig. 11. The generally very good agreement between the data sets gives a high degree of confidence that the approximations made in the CALIOP analyses have a relatively small impact on the quality of the CALIOP retrievals.

When there are several different layers within a column, the use of an incorrect lidar ratio in layers located higher in the atmosphere results in an error in the retrieved two-way transmittance of that layer. As that transmittance is used to rescale (renormalize) the input profile data used in the retrievals in lower layers, the renormalization error results in retrieval errors in these lower layers (Y1316). For example, if the lidar ratio used for analyzing an upper layer is too low, the retrieved two-way transmittance will be too high. As the attenuated backscatter profiles in underlying regions are divided by this two-way transmittance, the resulting renormalized lower layers will be too weak and the retrievals in these layers too low. On the other hand, the use of too high a lidar ratio in upper layers leads to retrievals that are too high in lower layers. These types of errors are cumulative and additional scrutiny should be applied to the optical properties retrieved from layers that underlie several other layers, or beneath appreciable optical depths.

The problem of selecting appropriate lidar ratios can be reduced by increasing the fraction of retrievals that are constrained by two-way transmittance measurements. The use of opaque water clouds (Hu et al., 2007) and/or the Earth's surface (Venkata and Reagan, 2016) as background targets against which the effective two-way transmittance of overlying layers can be estimated is expected to lead to further improvements in extinction retrievals in future versions of the CALIOP data products. But while useful and highly desirable, these full column constraints will not be a panacea, as the information contained in a single constraint must be carefully and correctly parsed out among all the layers detected within a single column.

Finally, we note that the V4 analysis algorithm architecture assumes that a single atmospheric feature detected at a single horizontal resolution has uniform optical properties throughout its vertical and horizontal extent. Any change in particulate type (e.g., from smoke to ice in pyrocumulonimbus) or in the phase of cloud particles (e.g. from randomly-oriented to horizontally-oriented ice, or from ice to water) will lead to errors in the retrieved profiles and optical depths. For example, the

lidar ratio for randomly oriented ice crystals is much higher than that for water droplets and so, in a mixed phase cloud with ice overlying water, the retrieval will most likely fail once the level at which the phase change is reached. The lidar ratio will then be reduced in steps until a successful retrieval is achieved, but the resulting solution will obviously be an artificially homogenized compromise between the correct values.

## 4 Summary and conclusions

This paper describes the changes made to the extinction and optical depth retrieval algorithms that are used to create the version 4.10 (V4) CALIOP level 2 data products. The extinction retrievals rely on inputs generated by several algorithms that execute earlier in the analysis process, and thus any V4 changes to these algorithms will also affect the retrieved extinction products to various degrees. Among the most significant of these are changes to the calibration procedures, improvements to the algorithms used for feature finding, cloud-aerosol discrimination, assignment of feature subtype, and extensive updates to the initial lidar ratios and multiple scattering factors. In this paper we concentrate on a description of changes to the extinction retrieval algorithm and the associated input parameters, and only discuss the changes to the other algorithms to the extent that they impact the extinction retrievals. Also, because the changes made to the V4 extinction algorithms mostly affect retrievals in clouds, where the extinction tends to be higher, while aerosol retrievals are more affected by changes to the preceding algorithms and input parameters, we have concentrated on reporting the former and only briefly summarize the changes expected in aerosol layers.

The main input parameters to the extinction retrieval process are the lidar ratios and multiple scattering factors for the particles in the features being analyzed. As discussed in Sect. 2.1.1 and presented in Table 1, there have been changes to the initial lidar ratios and their uncertainties for several of the aerosol types. In particular, there have been increases to the 532 nm lidar ratios for the clean marine (15 %), desert dust (10 %), and clean continental (51 %) aerosol types. In addition, a new dusty marine aerosol type has been added with a lidar ratio that is significantly higher than the clean marine type, but significantly lower than the polluted dust type. Absent any other changes, these higher lidar ratios will lead to higher values of retrieved particulate extinction. Similarly, reductions in the relative uncertainties ascribed to these lidar ratios will reduce the relative uncertainties in the retrieved extinction coefficients and optical depths. The new values of the aerosol lidar ratios in V4 have most often been derived from careful comparisons of CALIOP retrievals to measurements made by airborne HSRL along the same ground track and at around the time of the CALIOP overpass (Rogers et al., 2014; Burton et al., 2013). Use of these improved lidar ratios in retrieving the V4 CALIOP retrievals will produce results that are more representative of actual conditions than in previous data releases.

Among the changes to the extinction retrieval algorithms are revised criteria for determining the acceptability of a retrieval at a particular range step (height), discussed in Sect. 2.2.2. The algebraic solution for backscatter uncertainty (supplementary material Sect. S1.2) used in V4 now indicates when the current lidar ratio is too large for a solution to exist when used with the current input signal. An excessive number of attempts at a retrieval at a given range is now also interpreted as the use of a

lidar ratio that is too large. In both cases, the lidar ratio is reduced and the retrieval restarted from the top of the feature. These changes are reflected in changes to the extinction QC flag values as discussed in Sect. 2.2.6.

In the previous versions of the extinction retrieval algorithms, when the lidar ratio was determined to be too high to permit a solution and required reduction, it was reduced in steps, initially of 1 %, then of 5 %. As the sensitivity of retrieved extinction profiles and optical depths increases rapidly as the optical depth increases (e.g. Y1316), a reduction of this magnitude may well be too large for features with high optical depths, especially those that are determined to be opaque. This concern was supported by a study of opaque ROI clouds detected in January 2010. In V4, the algorithm for reducing the lidar ratio to achieve a solution has been changed so that it is smaller in optically thick features and larger in more tenuous features. (See Sect. 2.2.4.)

The most significant change to the extinction retrieval algorithms has been the introduction of a new algorithm for retrievals in opaque features discussed in Sect. 2.2.3. Whereas previous versions used a fixed value of lidar ratio for each feature type, the initial lidar ratio in V4 is now derived from CALIOP measurements, namely the integral of the layer attenuated backscatter signal. Should this value be too large to permit a retrieval, V4 implements a very conservative lidar ratio reduction scheme that produces more accurate extinction profiles. In particular, the large underestimates that frequently occurred in V3 have been eliminated (e.g., see Fig. 9).

It is quite difficult to verify these improvements by comparing with observations from other instruments. The lidar signal does not sample the whole depth of opaque clouds so cannot produce a full layer optical depth that could be compared (after scaling for wavelength differences) with passive instruments, which only produce an optical depth and not an extinction profile that could be compared with CALIOP's. Also, because the range of particle sizes in clouds, particularly near cloud tops, can span both the Rayleigh and Mie scattering ranges for radars, whereas the particles are very large compared with CALIOP's wavelength and are totally within the Mie scattering range, the shapes of CALIOP's extinction profiles are likely to be different from those retrieved by the Cloud Profiling Radar onboard the CloudSat satellite. Despite this lack of independent verification, evidence that the new opaque layer algorithm, with its new initialization of the lidar ratio and refined lidar ratio reduction scheme, has indeed led to improved retrievals was provided in Sect. 3.2.1 by a comparison of V4 and V3 retrievals of lidar ratios in the topmost ROI cloud layers detected at night during January 2010.

The newly developed surface detection scheme implemented in V4 markedly improves CALIOP's surface detection performance, and thus permits the use of the opaque layer algorithm with considerable confidence. Using layer integrated attenuated backscatter to calculate the initial lidar ratio in a feature that is wrongly identified as being opaque will result in lidar ratios that are too high with the result that an inaccurate extinction profile could be retrieved, depending on whether or not a lidar ratio reduction was triggered. The new surface detection algorithm ensures that such errors occur much less frequently than in the V3 data set, especially at nighttime when the signal-to-noise ratio is high and both the surface and the full vertical extent of the feature can be determined with high accuracy.

A second major improvement in the retrieval of extinction profiles and optical depths in semi-transparent ice clouds was achieved by incorporating a significant advance in the parameterization of ice cloud multiple scattering obtained from in-depth statistical analyses of several years of CALIPSO measurements. Using an extensive data set of CALIOP measurements of semi-transparent clouds collocated with CALIPSO imaging infrared radiometer optical depth retrievals, Garnier et al. (2015)

derived temperature-dependent multiple scattering factors and a corresponding temperature-dependent parameterization of initial cirrus lidar ratios. These new characterizations of light scattering in cirrus clouds have led to significant changes to the magnitude and height dependence of the extinction profiles in these clouds, as shown in Sect. 3.4. Sect. 3.3.2 presents a comparison of the optical depths reported in the MODIS collection 6 and CALIOP V4 data products for collocated semi-transparent ice clouds measured over the ocean during the day. The excellent agreement shown in Fig. 11 demonstrates the

substantial improvements made by adopting the new lidar ratios and multiple scattering factors employed in the V4 retrieval algorithms and gives confidence in the new analysis scheme.

**Data availability**

The CALIPSO lidar data products are available at the Atmospheric Science Data Center at NASA LaRC (https://eosweb.larc.nasa.gov/project/calipso/calipso_table) and at the AERIS/ICARE Data and Services Center

(http://www.icare.univ-lille1.fr). The CALIPSO+MODIS CALTRACK data are available solely from the AERIS/ICARE Data and Services Center.

**Supplement link**

**Author Contributions**

Stuart Young and Mark Vaughan developed the extinction retrieval algorithms and led the program of test runs of the various development versions with considerable assistance from Jim Lambeth.

The extensive suite of test data, which was designed to simulate actual CALIOP data expected from various types of atmospheric scenes and was used for developing and testing the extinction algorithms, was prepared by Kathy Powell.

The new temperature-dependent multiple scattering factor and lidar ratios for cirrus clouds were developed by Anne Garnier

and Mark Vaughan.

Data analysis for the paper was performed by Mark Vaughan, Jason Tackett and Stuart Young, who also prepared the figures with contributions from Anne Garnier.

Stuart Young led the writing of the manuscript with considerable contributions from Mark Vaughan and Anne Garnier.

**Competing interests**

The authors declare that they have no conflicts of interest.

**Acknowledgements**

The work described in this paper was supported by NASA Langley Research Center Contract NNL16AA05C (STARSS-III).
The authors thank Jay Kar, whose close examination of our manuscript helped us eliminate some subtle, but important, misstatements.

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
