# Peer review of "Extinction and Optical Depth Retrievals for CALIPSO's Version 4 Data Release"

_Atmospheric Measurement Techniques, 2018_

## Referee Comment (RC3) · Anonymous Referee #3 · 13 Jul 2018

I can only echo the findings of the other two reviewers. This paper is well organized and well written and certainly of interest to the broad cloud/radiation/remote sensing communities.

I do have one additional point to make though. It concerns the constrained retrieval in ice clouds. I am familiar with a the following paper: "Reverdy, et al. (2015), An EarthCARE/ATLID simulator to evaluate cloud description in climate models, J. Geophys. Res. Atmos., 120, 11,090–11,113, doi:10.1002/2015JD023919". If one looks in the appendix of this paper, there are some observations and lidar Monte-Carlo calculations that suggest that for small particle semi-transparent cirrus that the Rayleigh return below cloud may suffer from (small but sometimes not-insignificant) multiple-scattering induced decaying tails.

[Figure]

Remembering the Reverdy paper made me realise that I was not able to find any discussion of how the below cloud return altitude range for the constrained retrieval procedure in this submission. Accordingly, I think the addition of a few lines somewhere describing this and the possible (but likely limited) effect of multiple-scattering tails would not be out of place in this paper.

―――――――――――――――――――

---

## Short Comment (SC1) · 1 Aug 2018

This paper is a detailed and well-written discussion of retrieval algorithms used to produce the CALIPSO Version 4 data products. Nevertheless, a few comments:

Page 10, line 28: In Section 2.2.3 there is a mention that for opaque aerosol layers multiple scattering is assumed to be negligible. This is not necessarily true, particularly for dust layers, and so represents a source of error in the retrieval. This point should be made clear.

Page 14, line 27: It is the high optical depth that causes an increase in multiple scattering, not the width of the forward diffraction peak. The width of the forward peak does, however, lead to much more frequent scattering at large angles than in cirrus,

and significant amounts of pulse stretching.

Page 21, line 14 seems to be referring to figures 3d and 3f rather than 4d and 4f.

The "Caveats" section on pages 38-39 is a great addition to the paper. The second paragraph points out that the sensitivity of the extinction retrieval to differences between the lidar ratio used and the true lidar ratio increases with optical depth. This is an important point which will probably not be sufficiently appreciated by many readers and deserves some additional detail. As shown in the CALIPSO ATBDs (written by these same authors), retrieving an optical depth of even 3 to an accuracy of 10% requires knowing the lidar ratio to 0.1%. Most of the integrated signal being used to estimate the lidar ratio comes from the first two optical depths, which thus provides little constraint on small changes in lidar ratio in deeper parts of the cloud. Thus, data users should carefully consider the level of uncertainty in retrievals at high optical depths. These uncertainties are likely larger than the uncertainties reported in the product, which are estimated based on the assumption that lidar ratio is uniform throughout the retrieved layer.

Page 39, line 8: Makes a good point, but "is composed of the same material" is probably better stated as "has uniform optical properties"

Finally, anonymous referee #3 suggests the possibility of "decaying tails" below clouds due to multiple scattering, seen in Monte Carlo simulations of lidar returns from the upcoming ATLID lidar. He suggests these decaying tails might impact constrained retrievals or retrievals of lower layers. Similar Monte Carlo simulations of the CALIOP return signals (having a significantly larger field of view which tends to wholly capture the forward diffraction peak of cirrus particles) shows an impact of multiple scattering which is constant with range rather than decaying. Referee #3 is correct, though, that potential impacts of this on CALIOP retrievals deserves some discussion.

---

## Author Comment (AC1) · 26 Aug 2018

**Responses to Referee #1**

Anonymous Referee #1 Received and published: 9 July 2018

A valuable study for the changes made to the extinction and optical depth retrieval algorithms that are used to create the version 4.10 CALIOP Level 2 data products. A very interesting comparison of CALIPSO V3 and V4 with MODIS collections was conducted.

Some technical corrections will perfect this paper:

page 5, Table 1, the categories for tropospheric aerosol subtypes refer only in V4. i.e. elevated smoke and not just smoke as in V3

We thank the referee for reminding us of the change in the nomenclature of some aerosol types and have modified section 2.1.1 as shown in red text. (For consistency, we have also replaced "biomass burning", which was used in some earlier documents, with "smoke".)

**2.1.1 Initial lidar ratios for aerosols**

The initial lidar ratios and their uncertainties for several of the aerosol subtypes have been revised for V4 (Kim et al., 2018). As can be seen in Table 1, V4 specifies larger 532 nm lidar ratios for dust (by 10%), clean marine (by 15%) and clean continental (by 51%) aerosols. These higher lidar ratios, which are consistent with improved knowledge gained over the past decade (Kim et al., 2018), contribute to higher values in the retrieved backscatter, extinction and optical depths than were reported in V3. Our better understanding of lidar ratio variability led to significant reductions in the uncertainties ascribed to the lidar ratios for marine, desert dust and biomass burning smoke aerosols, resulting in generally in lower uncertainties in the retrieved products for these aerosol types. As explained in Kim et al. (2018) and documented in the V4 CALIPSO Data Products Catalog (https://www-calipso.larc.nasa.gov/documents/dpc\_index.php), the nomenclatures of the aerosol type 3 "polluted continental" and type 6 "smoke" were changed in version 4.1 to "polluted continental/smoke" and "elevated smoke"

respectively. V4 also adds a new "dusty marine" tropospheric subtype and five new stratospheric subtypes. The lidar ratio for the dusty marine mixture is significantly higher than that for clean marine, but significantly lower than that for polluted dust. (Aerosols identified as dusty marine in V4 were generally classified as polluted dust in V3.) Some aerosol lidar ratios at 1064 nm have also been changed, as shown in Table 1. Note that there was no cloud-aerosol classification in the stratosphere in V3 and that all stratospheric layers were assigned an initial lidar ratio of 25 sr and a multiple scattering factor of 1.

**page 7, Figure 1, the optical quality of this figure should be increased**

In recreating this figure, in addition to increasing the optical quality as requested, we have swapped panels (b) and (c) to agree with modified text in Section 2.1.2.1 and changed the axis labels to agree with AMT style guide.

**page 12, Figure 2, the axis labels and ticks should be increased in the first figure**

The figure has been recreated to increase the size of the ticks and the text.

page 13, line 10, "...which of these error sources cause..." Done

page 33, Figure 11a, (a) white symbol should be deleted Done

---

## Author Comment (AC4) · 26 Aug 2018

**Responses to short comment by David Winker.**

**D. Winker** david.m.winker@nasa.gov Received and published: 1 August 2018

This paper is a detailed and well-written discussion of retrieval algorithms used to produce the CALIPSO Version 4 data products.

Nevertheless, a few comments:

Page 10, line 28: In Section 2.2.3 there is a mention that for opaque aerosol layers multiple scattering is assumed to be negligible. This is not necessarily true, particularly for dust layers, and so represents a source of error in the retrieval. This point should be made clear.

The reviewer is correct: assuming that multiple scattering in opaque aerosol layers is negligible can indeed lead to retrieval errors. The additional text shown below in red explicitly acknowledges this.

For aerosols, multiple scattering is assumed to be negligible, and hence  $\eta$  is fixed at 1.0. However, this assumption may not be valid for opaque aerosol layers, and must be considered especially tenuous for opaque dust layers (Wandinger et al., 2010; Liu et al., 2011). In these cases, the  $\eta = 1$  assumption introduces bias errors similar to those incurred by the incorrect specification of lidar ratio (e.g. Y1316). Fortunately, because opaque aerosol layers occur only infrequently (~1% of all unique aerosol layers (and ~0.2% of all unique layers) detected in 2012 were identified as being opaque), the effects of these bias errors are somewhat mitigated.

**References**

Wandinger, U., M. Tesche, P. Seifert, A. Ansmann, D. Müller, and D. Althausen, 2010: "Size matters: Influence of multiple scattering on CALIPSO light-extinction profiling in desert dust", Geophys. Res. Lett., **37**, L10801, doi:10.1029/2010GL042815.

Liu, Z., D. Winker, A. Omar, M. Vaughan, C. Trepte, Y. Hu, K. Powell, W. Sun, B. Lin, 2011: "Effective lidar ratios of dense dust layers over North Africa derived from the CALIOP measurements", JQSRT, **112**, 204–213, doi:10.1016/j.jqsrt.2010.05.006.

Page 14, line 27: It is the high optical depth that causes an increase in multiple scattering, not the width of the forward diffraction peak. The width of the forward peak does, however, lead to much more frequent scattering at large angles than in cirrus, and significant amounts of pulse stretching.

We agree that our description of the cause of the range ambiguity in water clouds was not well explained. We trust that our revision is both clearer and more accurate.

The rules described above do not apply for lidar ratio uncertainties for opaque water clouds because of lidar "pulse stretching". Pulse stretching is caused by the broad forward peak of the scattering phase function of the small spherical droplets typically found in water clouds. The broad forward peak causes an increase in off-axis scattering in water clouds. With the increased multiple scattering found at high optical depths (e.g., in opaque water clouds), this off-axis scattering quickly dominates the backscattered signal with increasing penetration into the layer. Since lidar range information is measured by the time delay of the backscattered signal, off-axis scattering causes errors in the measurements.

**Page 21, line 14 seems to be referring to figures 3d and 3f rather than 4d and 4f.**

We thank the reviewer for pointing this out. The text now refers to Figure 3.

The "Caveats" section on pages 38-39 is a great addition to the paper. The second paragraph points out that the sensitivity of the extinction retrieval to differences between the lidar ratio used and the true lidar ratio increases with optical depth. This is an important point which will probably not be sufficiently appreciated by many readers and deserves some additional detail. As shown in the CALIPSO ATBDs (written by these same authors), retrieving an optical depth of even 3 to an accuracy of 10% requires knowing the lidar ratio to 0.1%. Most of the integrated signal being used to estimate the lidar ratio comes from the first two optical depths, which thus provides little constraint on small changes in lidar ratio in deeper parts of the cloud. Thus, data users should carefully consider the level of uncertainty in retrievals at high optical depths. These uncertainties are likely larger than the uncertainties reported in the product, which are estimated based on the assumption that lidar ratio is uniform throughout the retrieved layer.

We thank the reviewer for emphasizing the increasing sensitivity of the retrievals to even small errors in the lidar ratio as the optical depth increases. While we discussed this to some degree in Sect. 2.2.3, we have now added more material there to emphasize further the point he is making. We have also pointed the reader to section S1.3 in the supplementary material where the potential inadequacies of standard uncertainty analyses are explained. The additional material is shown in red below.

As a result, within any single layer, the degree to which the lidar ratio used in a retrieval differs from the actual value has a direct impact on the accuracy of the retrieved extinction coefficients and optical depths and this sensitivity increases with the optical depth of the feature (Y1316). For low optical depths, the relative error in the retrieved optical depth is closely approximated by the relative error in the lidar ratio, but for higher optical depths, the optical depth error increases by a factor of approximately ( $\exp(\tau) - 1$ ; See Eq. 40 to 43 in Young et al., 2013). As a consequence, even for a cloud with an optical depth as low as 3, in order to retrieve that optical depth to an accuracy of 10%, it is necessary to specify the lidar ratio to an accuracy of 0.1%. Note, too, that while the estimated uncertainties (section S1.2 in the Supplementary material) also increase with depth of penetration and increasing retrieved optical depth, they may underestimate the true uncertainties when the uncertainties in the input parameters are large and, therefore, do not adequately approximate the retrieval errors (see section S1.3 in the supplementary material).

Regarding his second point on the potential errors in a lidar ratio calculated from the integrated backscatter signal, we feel that we have already discussed this adequately on page 23, lines 5 - 13; page 23, line 18-19; and on page 41, line 10.

His third point, about the errors arising from the assumption of a constant lidar ratio throughout a feature, is discussed in the final paragraph of the Caveats section.

Page 39, line 8: Makes a good point, but "is composed of the same material" is probably better stated as "has uniform optical properties"

Changed as suggested.

Finally, anonymous referee #3 suggests the possibility of "decaying tails" below clouds due to multiple scattering, seen in Monte Carlo simulations of lidar returns from the upcoming ATLID lidar. He suggests these decaying tails might impact constrained retrievals or retrievals of lower layers. Similar Monte

Carlo simulations of the CALIOP return signals (having a significantly larger field of view which tends to wholly capture the forward diffraction peak of cirrus particles) shows an impact of multiple scattering which is constant with range rather than decaying. Referee #3 is correct, though, that potential impacts of this on CALIOP retrievals deserves some discussion.

We have discussed the likely frequency of occurrence and potential impacts of such "decaying tails" on CALIOP's constrained retrievals in our response to Referee #3, where we also indicate the additions we have made to Sections 2.2.1 and the Caveats in Section 3.6.

---

## Author Comment (AC3)

**Responses to Referee # 3**

Anonymous Referee #3

I can only echo the findings of the other two reviewers. This paper is well organized and well written and certainly of interest to the broad cloud/radiation/remote sensing communities.

I do have one additional point to make though. It concerns the constrained retrieval in ice clouds. I am familiar with a the following paper: "Reverdy, et al. (2015), An EarthCARE/ATLID simulator to evaluate cloud description in climate models, J. Geophys.Res. Atmos., 120, 11,090–11,113, doi:10.1002/2015JD023919". If one looks in the appendix of this paper, there are some observations and lidar Monte-Carlo calculations that suggest that for small particle semi-transparent cirrus that the Rayleigh return below cloud may suffer from (small but sometimes not-insignificant) multiple-scattering induced decaying tails.

Remembering the Reverdy paper made me realise that I was not able to find any discussion of how the below cloud return altitude range [is determined?] for the constrained retrieval procedure in this submission. Accordingly, I think the addition of a few lines somewhere describing this and the possible (but likely limited) effect of multiple-scattering tails would not be out of place in this paper.

We certainly agree that it is possible that there may occasionally be some conditions in which our parameterization of multiple scattering in cirrus clouds causes biases in constrained retrievals of optical depth. We have added material, shown in red, to Section 2.2.1 to describe how feature boundaries are refined and how clear air regions are identified.

Constrained retrievals use measurements of the effective two-way layer transmittance and its uncertainty that are determined by comparing signals from above and below the layer. The determination of the layer boundaries is detailed in Vaughan et al. (2005 and 2009). Briefly, the initial determination of cloud base is determined as that range at which the attenuated scattering ratio (the ratio of the normalized, range-corrected backscatter signal to a molecular backscatter model) drops below a range-dependent threshold that is determined largely by SNR and signal attenuation by the overlying atmosphere. This initial estimate of cloud base is further refined by continuing to search below this altitude for a region where the attenuated scattering ratio ceases to be a decreasing function of range. Depending on the SNR in the region below the cloud, this refinement sets cloud base below any readily detectable "leakage" of the signal into the assumed clear region caused by, for example, the transient recovery time of the detectors or by multiple scattering from small particles in dense clouds. Note that this multiple scattering leakage is not pulse stretching (Miller and Stephens, 1999), but instead occurs for instance because the forward scattering angle from the small ice crystals in cold cirrus can be wider than the CALIOP receiver field of view (Reverdy et al., 2015), so that the multiple scattering factor can be slightly larger below the ice clouds than in cloud (Winker et al, 2003). Once all features have been detected in a column, so called "clear air" regions above and below each feature are identified. To initiate a constrained retrieval, the V4 CALIOP extinction algorithm requires a minimum feature-free vertical extent of 2.48 km both above and below a candidate feature. The required effective layer two-way transmittance is then calculated as the ratio of the mean attenuated scattering ratios computed over these below-cloud and above-cloud clear air regions. The fidelity of these estimates relies on the supposition that the backscatter signals in the clear air regions are due solely to air molecules. If this condition is not met (and this is impossible to confirm with absolute certainty), then unless the mean particulate scattering ratios in the two clear air regions are identical, the transmittance measurements will be in error, no matter how small the reported uncertainty, and the constrained retrieval will also be in error. These are bias errors, not random errors, as discussed in Sect. 3b2 of Young et al. (2013), and by del Guasta (1998). Undetected particulate layers above the layer being analyzed can also affect the calculated lidar ratio. Extreme errors can cause the derived lidar ratios to approach and sometimes even exceed the physically acceptable limits of 0.05 sr to 250 sr imposed by the V4 retrieval scheme. Any constrained retrieval (bit 0 set to 1 in the extinction QC flag –

See Sect. 2.2.6) in which the derived lidar ratio is equal to either of these limits is to be treated as suspect. In these cases, the lidar ratio is set to the limit and an unconstrained retrieval is performed using this value. Such cases are indicated by the setting of bit 8 to 1 in the extinction QC flag (see Sect. 2.2.6 and Table 2), giving a total value of 257.

We have also added the following material after the current second paragraph of Section 3.6 (Caveats) and suggest that constrained retrievals of optical depth and lidar ratio be compared with adjacent unconstrained values if multiple scattering biases are suspected. Like Reverdy et al. (2015), we consider a detailed discussion of the intricacies of multiple scattering and its effects and treatment well beyond the scope of this paper.

There are some circumstances in which constrained retrievals may also be in error. As explained in Sect. 2.2.1, particulate scattering in regions used for normalization can lead to biased results. Also, as discussed by Reverdy et al. (2015), forward scattering from small ice crystals within a cloud can cause an enhancement in the backscatter signal measured below the cloud that decreases with range below cloud base. They further suggest that, for CALIOP signals, the rate of decay is so long that it is only really notable for cirrus that are composed of small particles (e.g., 10 μm – 20 μm) and have relatively high optical depths. When applying the CALIOP two-way transmittance estimation algorithm, the impact of such an enhancement would be to produce a constraint that is biased high, with the result that both the retrieved optical depth and lidar ratio would be biased low. However, these conditions occur relatively infrequently. For all V4 constrained retrievals of ice cloud profiles measured between 60° S and 60° N during the years 2011 – 2015, only 2.1 % have both centroid temperatures below –70 °C, where particles can be small (Heymsfield et al., 2014), and optical depths greater than 0.5.

If multiple scattering induced biases are suspected, the lidar ratios and optical depths where constrained retrievals are employed should be compared with the same parameters in adjacent columns that use unconstrained retrievals. In any case, constrained retrievals in which lidar ratios and optical depths have high relative uncertainties should also be regarded with caution. Finally, in order to assess the likely impact of these potential errors, we refer the reader to the comparison of CALIOP V4 and MODIS C6 optical depths presented in Section 3.3.2 and in Fig. 11. The generally very good agreement between the data sets gives a high degree of confidence that the approximations made in the CALIOP analyses have a relatively small impact on the quality of the CALIOP retrievals.

**References**

Heymsfield, A., Winker, D., Avery, M., Vaughan, M., Diskin, G., Deng, M., Mitev, V., and Matthey, R.: Relationships between ice water content and volume extinction coefficient from in situ observations for temperatures from 0° to -86 °C: Implications for spaceborne lidar retrievals, J. Appl. Meteorol. Clim., 53, 479–505, doi:10.1175/JAMC-D-13-087.1, 2014.

Miller, S. D. and Stephens, G. L.: Multiple scattering effects in the lidar pulse stretching problem, J. Geophys. Res., 104, 22205–22219, doi:10.1029/1999JD900481, 1999.

Winker, D. M.: Accounting for multiple scattering in retrievals from space lidar, Proc. SPIE Int. Soc. Opt. Eng., 5059, 128–139, 2003.

---

## Author Comment (AC5)

Since posting our initial response, a sharp-eyed internal reviewer has advised us that the last two sentences in our revisions for section 2.2.1 do not accurately describe the error-handling procedures implemented in the CALIOP version 4 extinction retrieval code. Consequently, we have further modified the text, both here in our updated response and in the revised manuscript.

The revised paragraph from our previous response is shown below. The text in orange was inserted as part of our initial response to the comments from referee #3. The text rendered in gray strike through was also included as part of this initial response, but has now been removed and replaced by the text in blue.

Constrained retrievals use measurements of the effective two-way layer transmittance and its uncertainty that are determined by comparing signals from above and below the layer. The determination of the layer boundaries is detailed in Vaughan et al. (2005 and 2009). Briefly, the initial determination of cloud base is determined as that range at which the attenuated scattering ratio (the ratio of the normalized, range-corrected backscatter signal to a molecular backscatter model) drops below a range-dependent threshold that is determined largely by SNR and signal attenuation by the overlying atmosphere. This initial estimate of cloud base is further refined by continuing to search below this altitude for a region where the attenuated scattering ratio ceases to be a decreasing function of range. Depending on the SNR in the region below the cloud, this refinement sets cloud base below any readily detectable "leakage" of the signal into the assumed clear region caused by, for example, the transient recovery time of the detectors or by multiple scattering from small particles in dense clouds. Note that this multiple scattering leakage is not pulse stretching (Miller and Stephens, 1999), but instead occurs for instance because the forward scattering angle from the small ice crystals in cold cirrus can be wider than the CALIOP receiver field of view (Reverdy et al., 2015), so that the multiple scattering factor can be slightly larger below the ice clouds than in cloud (Winker et al, 2003). Once all features have been detected in a column, so called "clear air" regions above and below each feature are identified. To initiate a constrained retrieval, the V4 CALIOP extinction algorithm requires a minimum feature-free vertical extent of 2.48 km both above and below a candidate feature. The required effective layer two-way transmittance is then calculated as the ratio of the mean attenuated scattering ratios computed over these below-cloud and above-cloud clear air regions. The fidelity of these estimates relies on the supposition that the backscatter signals in the clear air regions are due solely to air molecules. If this condition is not met (and this is impossible to confirm with absolute certainty), then unless the mean particulate scattering ratios in the two clear air regions are identical, the transmittance measurements will be in error, no matter how small the reported uncertainty, and the constrained retrieval will also be in error. These are bias errors, not random errors, as discussed in Sect. 3b2 of Young et al. (2013), and by del Guasta (1998). Undetected particulate layers above the layer being analyzed can also affect the calculated lidar ratio. Extreme errors can cause the derived lidar ratios to approach and sometimes even exceed the physically acceptable limits of 0.05 sr to 250 sr imposed by the V4 retrieval scheme. Any constrained retrieval (bit 0 set to 1 in the extinction QC flag - See Sect. 2.2.6) in which the derived lidar ratio is equal to either of these limits is to be treated as suspect. In these cases, the lidar ratio is set to the limit and an unconstrained retrieval is performed using this value. Such cases are indicated by the setting of bit 8 to 1 in the extinction QC flag (see Sect. 2.2.6 and Table 2), giving a total value of 257. When serious errors are encountered in the solution of constrained retrievals, bit 8 is also set, giving an extinction QC flag value of 257.